# Can Learned Optimization Make Reinforcement Learning Less Difficult?

**Alexander D. Goldie**[*12]    **Chris Lu**[1]    **Matthew T. Jackson**[12]
**Shimon Whiteson**[†2]    **Jakob N. Foerster**[†1]
[1]FLAIR, University of Oxford    [2]WhiRL, University of Oxford

## Abstract

While reinforcement learning (RL) holds great potential for decision making in the real world, it suffers from a number of unique difficulties which often need specific consideration. In particular: it is highly non-stationary; suffers from high degrees of plasticity loss; and requires exploration to prevent premature convergence to local optima and maximize return. In this paper, we consider whether learned optimization can help overcome these problems. Our method, Learned **O**ptimization for **P**lasticity, **E**xploration and **N**on-stationarity (OPEN[1]), meta-learns an update rule whose input features and output structure are informed by previously proposed solutions to these difficulties. We show that our parameterization is flexible enough to enable meta-learning in diverse learning contexts, including the ability to use stochasticity for exploration. Our experiments demonstrate that when meta-trained on single and small sets of environments, OPEN outperforms or equals traditionally used optimizers. Furthermore, OPEN shows strong generalization characteristics across a range of environments and agent architectures.

## 1    Introduction

Reinforcement learning [1, RL] has undergone significant advances in recent years, scaling from solving complex games [2, 3] towards approaching real world applications [4–7]. However, RL is limited by a number of difficulties which are not present in other machine learning domains, requiring the development of numerous hand-crafted workarounds to maximize its performance.

Here, we take inspiration from three *difficulties* of RL: non-stationarity due to continuously changing input and output distributions [8]; high degrees of plasticity loss limiting model capacities [9, 10]; and exploration, which is needed to ensure an agent does not converge to local optima prematurely [1, 11]. Overcoming these challenges could enable drastic improvements in the performance of RL, potentially reducing the barriers to applications of RL in the real-world. Thus far, approaches to tackle these problems have relied on human intuition to find *hand-crafted* solutions. However, this is fundamentally limited by human understanding. *Meta-RL* [12] offers an alternative in which RL algorithms themselves are learned from data rather than designed by hand. Meta-learned RL algorithms have previously demonstrated improved performance over hand-crafted ones [13–15].

On a related note, *learned optimization* has proven successful in supervised and unsupervised learning (e.g., VeLO [16]). Learned optimizers are generally parameterized update rules trained to outperform handcrafted algorithms like gradient descent. However, current learned optimizers perform poorly in RL [16, 17]. While some may argue that RL is simply an out-of-distribution task [16], the lack of

---

[*]Correspondence to goldie@robots.ox.ac.uk.

[†]Equal supervision.

[1]Open-source code is available here.

consideration for, and inability to *target*, specific difficulties of RL in naïve learned optimizers leads us to believe that they would still underperform even if RL were in the training distribution.

We propose an algorithm, shown in Figure 1, for meta-learning optimizers specifically for RL called Learned **O**ptimization for **P**lasticity, **E**xploration and **N**onstationarity (OPEN). OPEN meta-learns update rules whose inputs are rooted in previously proposed solutions to the difficulties above, and whose outputs use *learnable stochasticity* to boost exploration. OPEN is trained to maximize final return *only*, meaning it does not use regularization to mitigate the difficulties it is designed around.

In our results (Section 6), we benchmark OPEN against handcrafted optimizers (Adam [18], RMSProp [19]), open-source discovered/learned optimizers (Lion [20], VeLO [16]) and baseline learned optimizers trained for RL (Optim4RL [17], '*No Features*'). We show that OPEN can fit to single and small sets of environments and also generalize in- and out-of-support of its training distribution. We further find that OPEN generalizes better than Adam [18] to gridworlds and Craftax-Classic [21]. Therefore, our novel approach of *increasing* optimizer flexibility via extra inputs and an exploration-driven output, rather than enforcing more structured updates, enables significant performance gains.

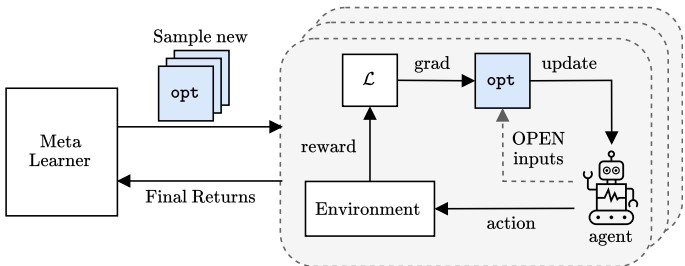

Figure 1: A visualization of OPEN. We train $N$ agents, replacing the handcrafted optimizer of the RL loop with ones sampled from the meta-learner (i.e., evolution). Each optimizer conditions on gradient, momentum and additional inputs, detailed in Section 5.3, to calculate updates. The final returns from each loop are output to the meta learner, which improves the optimizer before repeating the process. A single inner loop step is described algorithmically in Appendix B.1.

## 2 Background

**Reinforcement Learning**   The RL problem is formulated as a *Markov Decision Process* [1, MDP] described by the tuple $\langle \mathcal{A}, \mathcal{S}, P, R, \rho, \gamma \rangle$. At discrete timestep $t$, the agent takes action $a_t \in \mathcal{A}$ sampled from its (possibly stochastic) policy, $\pi(\cdot|s_t) \in \Pi$, which conditions on the current state $s_t \in \mathcal{S}$ (where $s_0 \sim \rho$). After each action, the agent receives reward $R(s_t, a_t)$ and the state transitions to $s_{t+1}$, based on the transition dynamics $P(s_{t+1}|s_t, a_t)$. An agent's objective is to maximize its *discounted expected return*, $J^\pi$, corresponding to a discount factor $\gamma \in [0, 1)$, which prevents the agent making *myopic* decisions. This is defined as

$$J^\pi := \mathbb{E}_{a_{0:\infty} \sim \pi, s_0 \sim \rho, s_{1:\infty} \sim P} \left[ \sum_{t=0}^{\infty} \gamma^t R_t \right]. \tag{1}$$

Proximal policy optimization [22, PPO] is an algorithm designed to maximize $J^\pi$. It uses advantage, $A^\pi(s, a)$, which is calculated from the *state value function*, $V^\pi = \mathbb{E}_\pi[\sum_{t=0}^{\infty} \gamma^t R_t | S_t = s]$, and *state-action value function*, $Q^\pi(s, a) = \mathbb{E}_\pi[\sum_{t=0}^{\infty} \gamma^t R_t | S_t = s, A_t = a]$. It measures the improvement of a specific action over the current policy and takes the form

$$A^\pi(s, a) = Q^\pi(s, a) - V^\pi(s). \tag{2}$$

PPO introduces a loss for optimizing the policy, parameterized by $\theta$, that prevents extreme policy updates in gradient *ascent*. This uses *clipping*, which ensures there is no benefit to updating $\theta$ beyond where the policy *probability ratio*, $r_t(\theta) = \frac{\pi_\theta(a_t|s_t)}{\pi_{\theta_{old}}(a_t|s_t)}$, exceeds the range $[1 \pm \epsilon]$. The clipped loss is

$$L^{CLIP}(\theta) = \mathbb{E}\left[ \min(r_t(\theta) A^\pi(s_t, a_t), \text{clip}(r_t(\theta), 1 \pm \epsilon) A^\pi(s_t, a_t)) \right]. \tag{3}$$

PPO is an actor-critic [1] method, where the policy and value functions are modeled with different neural networks, or separate heads of the same neural network, conditioning on state. The PPO objective to maximize combines the clipped loss, a value function error, and an entropy bonus into

$$L_t(\theta) = \mathbb{E}[L_t^{CLIP}(\theta) - c_1 L_t^{VF}(\theta) + c_2 S[\pi_\theta](s_t)]. \tag{4}$$

**Meta-Learning Optimizers in RL**    Algorithms like Adam [18] or RMSprop [19] are designed to maximize an objective by updating the parameters of a neural network in the direction of *positive gradient* with respect to the function. They are often applied with augmentations such as *momentum* or *learning rate schedules* to better converge to optima. Learned optimizers offer an alternative: parameterized *update rules*, conditioning on more than just gradient, which are trained to maximize an objective [23, 16, 24]. For any parameterized optimizer $\text{opt}_\phi$, which conditions on a set of inputs $\boldsymbol{x}$, the update rule to produce update $u$ can be described as a function, $\text{opt}_\phi(\boldsymbol{x}) \to u$.

We treat learning to optimize as a meta-RL problem [12]. In meta-RL, the goal is to maximize $J^\pi$ over a *distribution* of MDPs $P(\mathcal{M})$. For our task, an optimizer trained to maximize $\mathcal{J}(\phi) = \mathbb{E}_\mathcal{M}\left[J^\pi|\text{opt}_\phi\right]$ yields the optimal meta-parameterization $\text{opt}_{\phi^*}$.

**Evolution Strategies**    Evolution algorithms (EA) are a *backpropagation-free*, black-box method for optimization [25] which uses a population of *perturbed* parameters sampled from a distribution (here, $\hat{\theta} \sim \mathcal{N}(\theta, \sigma^2 I)$). This population is used to maximize a *fitness* $F(\cdot)$. EA encompasses a range of techniques (e.g., evolution strategies (ES) [26, 27], genetic algorithms [28] or CMA-ES [29]).

Natural evolution strategies (NES) [30] are a class of ES methods that use the population fitness to estimate a natural gradient for the mean parameters, $\theta$. This can then be optimized with typical gradient ascent algorithms like Adam [18]. Salimans et al. [26] introduce *OpenAI ES* for optimizing $\theta$ using the estimator

$$\nabla_\theta \mathbb{E}_{\epsilon \sim N(0,I)} F(\theta + \sigma\epsilon) = \frac{1}{\sigma} \mathbb{E}_{\epsilon \sim N(0,I)} \{F(\theta + \sigma\epsilon)\epsilon\}, \tag{5}$$

which is approximated using a population average. In practice, we use antithetic sampling (i.e., for each sampled $\epsilon$, evaluating $+\epsilon$ and $-\epsilon$) [31]. Antithetic *task* sampling enables learning on a task *distribution*, by evaluating and ranking each antithetic pair on *different* tasks [32].

Historically, RL was too slow for ES to be practical for meta-training. However, PureJaxRL [15] recently demonstrated the feasibility of ES for meta-RL, owing to the speedup enabled by vectorization in Jax [33]. We use the implementation of *OpenAI ES* [26] from evosax [34].

# 3    Related Work

## 3.1    Optimization in RL

Wilson et al. [35] and Reddi et al. [36] show that adaptive optimizers struggle in highly stochastic processes. Henderson et al. [37] indicate that, unlike other learning regimes, RL is sufficiently stochastic for these findings to apply, which suggests RL-specific optimizers could be beneficial.

The idea that optimizers designed for supervised learning may not perfectly transfer to RL is reflected by Bengio et al. [38], who propose an amended momentum suitable for temporal difference learning [39]. This is related to work by Sarigül and Avci [40], who explore the impact of different types of momentum on RL. While these works motivate designing optimization techniques *specifically* for RL, we take a more expressive approach by replacing the whole optimizer instead of just its momentum calculation, and using meta-learning to fit our optimizer to data rather than relying on potentially suboptimal human intuition.

## 3.2    Meta-learning

**Discovering RL Algorithms**    Rather than using handcrafted algorithms, a recent objective in meta-RL is *discovering* RL algorithms from data. While there are many successes in this area (e.g., Learned Policy Gradient [13, LPG], MetaGenRL [14], Active Adaptive Perception [41] and Learned Policy Optimisation [15, LPO]), we focus on meta-learning a replacement to the *optimizer* due to the outsized impact a learned update rule can have on learning. We also use *specific* difficulties of RL to guide the *design* of our method, rather than simply applying end-to-end learning.

Jackson et al. [32] learn temporally-aware versions of LPO and LPG. While their approach offers inspiration for dealing with non-stationarity in RL, they also rely on Adam [18], an optimizer designed for stationarity that is suboptimal for RL [9]. Instead, we propose replacing the *optimizer* itself with an expressive and dynamic update rule that is not subject to these problems.

**Learned Optimization**   Learning to optimize [23, 42, 43, L2O] strives to learn replacements to handcrafted gradient-based optimizers like Adam [18], generally using neural networks (e.g., [24, 44–46]). While they show strong performance in supervised and unsupervised learning [16, 47], previous learned optimizers do not consider the innate difficulties of RL, limiting transferability. Furthermore, while VeLO used 4000 TPU-months of compute [16], OPEN requires on the order of one GPU-month.

Lion [20] is a symbolic optimizer discovered by evolution that outperforms AdamW [48] using a similar update expression. While the simplistic, symbolic form of Lion lets it generalize to RL better than most learned optimizers, it cannot condition on features additional to gradient and parameter value. This limits its expressibility, ability to target the difficulties of RL and, therefore, performance.

**Learned Optimization in RL**   Learned optimization in RL is significantly more difficult than in other learning domains due to the problems outlined in section 4. For this reason, SOTA learned optimizers fail in transfer to RL [16]. Optim4RL [17] attempts to solve this issue by learning to optimize directly in RL. However, in their tests and ours, Optim4RL fails to consistently beat handcrafted benchmarks. Also, it relies on a heavily constrained update expression based on Adam [18], and it needs expensive learning rate tuning. Instead, we achieve much stronger results with a completely *black-box* setup inspired by preexisting methods for mitigating specific difficulties in RL.

# 4   Difficulties in RL

We believe that fundamental differences exist between RL and other learning paradigms which make RL particularly difficult. Here, we briefly cover a specific set of prominent difficulties in RL, which are detailed with additional references in appendix A. Our method takes inspiration from handcrafted heuristics targeting these challenges (Section 5.3). We show via thorough ablation (Section 7) that explicitly formulating our method around these difficulties leads to significant performance gains.

**(Problem 1) Non-stationarity**   RL is subject to non-stationarity over the training process [1] as the updating agent causes changes to the training distribution. We denote this *training non-stationarity*. Lyle et al. [9] suggest optimizers designed for stationary settings struggle under nonstationarity.

**(Problem 2) Plasticity loss**   Plasticity loss, or the inability to fit new objectives during training, has been a theme in recent deep RL literature [9, 49, 50, 10]. Here, we focus on dormancy [49], a measurement tracking inactive neurons used as a metric for plasticity loss [10, 51–53]. It is defined as

$$s_i^l = \frac{\mathbb{E}_{x \in D}|h_i^l(x)|}{\frac{1}{H^l}\sum_{k \in h}\mathbb{E}_{x \in D}|h_k^l(x)|}, \tag{6}$$

where $h_i^l(x)$ is the activation of neuron $i$ in layer $l$ with input $x \in D$ for distribution $D$. $H^l$ is the total number of neurons in layer $l$. The denominator normalizes average dormancy to 1 in each layer.

A neuron is $\tau$-dormant if $s_i^l \leq \tau$, meaning the neuron's output makes up less than $\tau$ of its layer's output. For ReLU activation functions, $\tau = 0$ means a neuron is in the saturated part of the ReLU. Sokar et al. [49] find that dormant neurons generally *stay* dormant throughout training, motivating approaches which try to reactivate dormant neurons to boost plasticity.

**(Problem 3) Exploration**   *Exploration* is a key problem in RL [1]. To prevent premature convergence to local optima, and thus maximize final return, an agent must explore uncertain states and actions. Here, we focus on parameter space noise for exploration [11], where noise is applied to the parameters of the agent rather than to its output actions, like $\epsilon$-greedy [1].

# 5   Method

There are three key considerations when doing learned optimization for RL: what architecture to use; how to train the optimizer; and what inputs the optimizer should condition on. In this section, we systematically consider each of these questions to construct OPEN, with justification for each of our decisions grounded in our core difficulties of RL.

## 5.1   Architecture and Parameterization

To enable conditioning on history, which is required to express behavior like momentum, for example, OPEN uses a gated recurrent unit [54, GRU]. This is followed by two fully connected layers with

LayerNorm [55], which we include for stability. All layers are small; this is important for limiting memory usage, since the GRU stores separate states for *every* agent parameter, and for maintaining computational efficiency [45]. We visualize and detail the architecture of OPEN in Appendix B.

**Update Expression**    We split the calculation of the update in OPEN into three stages, each of which serves a different purpose. The first stage, which follows Metz et al. [24], is

$$\hat{u}_i = \alpha_1 m_i \exp \alpha_2 e_i, \tag{7}$$

where $m_i$ and $e_i$ are optimizer outputs for parameter $i$, and $\alpha_{(1,2)}$ are small scaling factors used for stability. This update can cover many orders of magnitude without requiring large network outputs.

**(P3)**    Secondly, we augment the update rule for the *actor* only to increase exploration; there is no need for the critic to explore. We take inspiration from parameter space noise for exploration [11], since it can easily be applied via the optimizer. To be precise, we augment the actor's update as

$$\hat{u}_i^{\text{actor, new}} := \hat{u}_i^{\text{actor}} + \alpha_3 \delta_i^{\text{actor}} \epsilon. \tag{8}$$

Here, $\alpha_3$ is a small, stabilizing scaling factor, $\delta_i^{\text{actor}}$ is a third output from the optimizer, and $\epsilon \sim \mathcal{N}(0,1)$ is sampled Gaussian noise. By multiplying $\delta_i^{\text{actor}}$ and $\epsilon$, we introduce a random walk of learned, per-parameter variance to the update which can be used for exploration. Since $\delta_i^{\text{actor}}$ depends on the optimizer's inputs, this can potentially learn complex interactions between the noise schedule and the features outlined in Section 5.3. Unlike Plappert et al. [11], who remove and resample noise between rollouts, OPEN adds permanent noise to the parameters. This benefits plasticity loss (i.e., **(P2)**), in principle enabling the optimizer to reactivate dormant neurons in the absence of gradient.

Unfortunately, naïve application of the above updates can cause errors as, even without stochasticity, agent parameters can grow to a point of numerical instability. This causes difficulty in domains with *continuous* action spaces, where action selection can involve exponentiation to get a non-negative standard deviation. Therefore, the *final* update stage stabilizes meta-optimization by zero-meaning, as

$$\mathbf{u} = \hat{\mathbf{u}} - \mathbb{E}[\hat{\mathbf{u}}]. \tag{9}$$

While this limits the update's expressiveness, we find that even traditional optimizers tend to produce nearly zero-mean updates. In practice, this enables learning in environments with continuous actions without harming performance for discrete actions. Parameter $i$ is updated as $p_i^{(t+1)} = p_i^{(t)} - u_i$.

## 5.2   Training

We train our optimizers with *OpenAI ES* [26], using *final return* as the fitness. We apply the commonly-used rank transform, which involves mapping rankings over the population to the range $[-0.5, 0.5]$, to the fitnesses before estimating the ES gradient; this is a form of fitness shaping [30, 26], and makes learning both easier and invariant to reward scale.

For training on multiple environments simultaneously (i.e., multi-task training, Section 6), we evaluate *every* member of the population on *all* environments. After evaluation, we: 1) Divide by the return Adam achieves in each environment; 2) Average the scores over environments; 3) Do a rank transform. Normalizing by Adam maps returns to a roughly common scale, enabling comparison between diverse environments. However, this biases learning to environments where Adam underperforms OPEN. We believe finding better curricula for multi-task training would be highly impactful future work.

## 5.3   Inputs

Carefully selecting which inputs to condition OPEN on is crucial; they should be sufficiently expressive without significantly increasing the computational cost or meta-learning sample requirements of the optimizer, and should allow the trained optimizers to surgically target specific problems in RL. To satisfy these requirements, we take inspiration from prior work addressing our focal difficulties of RL. In spirit, we distill current methods to a 'lowest common denominator' which is cheap to calculate. We provide additional details of how these features are calculated in Appendix B.2.

**(P1) Two training timescales**    Many learned optimizers for stationary problems incorporate some version of progress through training as an input [16, 45, 46]. Since PPO learns from successively collected, stationary batches of data [56], we condition OPEN on how far it is through updating with

the current batch (i.e., *batch proportion*). This enables behavior like learning rate scheduling, which has proved effective in stationary problems (e.g., [57, 58]), and bias correction from Adam to account for inaccurate momentum estimates [18].

Inspired by Jackson et al. [32], who demonstrate the efficacy of learning dynamic versions of LPG [13] and LPO [15], we also condition OPEN on how far the current batch is through the total number of batches to be collected (i.e., *training proportion*). This directly targets *training non-stationarity*.

**(P2) Layer Proportion**    Nikishin et al. [59, 60] operate on higher (i.e., closer to the output) network layers in their attempts to address plasticity loss in RL. Furthermore, they treat intervention depth as a hyperparameter. To replicate this, and enable varied behavior between the different layers of an agent's network, we condition OPEN on the relative position of the parameter's layer in the network.

**(P2) Dormancy**    As Sokar et al. [49] *reinitialize $\tau$-dormant* neurons, we condition OPEN *directly* on dormancy; this enables similar behavior by allowing OPEN to react as neurons become more dormant. In fact, in tandem with learnable stochasticity, this enables OPEN to reinitialize dormant neurons, just like Sokar et al. [49]. Since dormancy is calculated for *neurons*, rather than *parameters*, we use the value of dormancy for the neuron *downstream* of each parameter.

# 6    Results

In this section, we benchmark OPEN against a plethora of baselines on large-scale training domains.

## 6.1    Experimental Setup

Due to computational constraints, we meta-train an optimizer on a single random seed without ES hyperparameter tuning. This follows standard evaluation protocols in learned optimization (e.g., [16, 17, 47]), which are also constrained by the high computational cost of meta-learned optimization. We provide the cost of experiments in Appendix J, including a comparison of runtimes with other optimizers. We detail hyperparameters in Appendix C.5. We define four evaluation domains, based on Kirk et al. [61], in which an effective learned optimization framework *should* prove competent:

**Single-Task Training**    A learned optimizer must be capable of fitting to a single environment, and being evaluated in the *same* environment, to demonstrate it is expressive enough to learn *an* update rule. We test this in five environments: Breakout, Asterix, Space Invaders and Freeway from MinAtar [62, 63]; and Ant from Brax [64, 65]. This is referred to as 'singleton' training in Kirk et al. [61].

**Multi-Task Training**    To show an optimizer is able to perform under a wide input distribution, it must be able to learn in a number of environments *simultaneously*. Therefore, we evaluate performance from training in all four environments from MinAtar [62, 63][2]. We evaluate the average normalized score across environments with respect to Adam.

**In-Distribution Task Generalization**    An optimizer should generalize to unseen tasks within its training distribution. To this end, we train OPEN on a *distribution* of gridworlds from Jackson et al. [66] with antithetic task sampling [32], and evaluate performance by sampling tasks from the *same distribution*. We include details in Appendix D.

**Out-Of-Support Task Generalization**    Crucially, an optimizer unable to generalize to new settings has limited real-world usefulness. Therefore, we explore *out-of-support* (OOS) generalization by testing OPEN on specific gridworld distributions defined by Oh et al. [13], and a set of mazes from minigrid [67] which are not in the training distribution, with unseen agent parameters. Furthermore, we test how OPEN performs *0-shot* in Craftax-Classic [21], a Jax reimplementation of Crafter [68].

**Baselines**    We compare against open-source implementations [69] of Adam [18], RMSProp [19], Lion [20] and VeLO [16, 45]. We also *learn* two optimizers for the *single-* and *multi*-task settings: '*No Features*', which only conditions on gradient and momentum; and Optim4RL [17] (using ES instead of meta-gradients, as in Lan et al. [17]). Since Optim4RL is initialized close to sgn(Adam), and tuning its learning rate is too practically expensive, we set a learning rate of $0.1 \times \text{LR}_{\text{Adam}}$ based on Lion [20] (which moves from AdamW to sgn(AdamW)). The optimizer's weights can be scaled to compensate if this is suboptimal. We primarily consider interquartile mean (IQM) of final return with 95% stratified bootstrap confidence intervals [70]. All hyperparameters can be found in Appendix C.

---

[2]Seaquest, which is a part of MinAtar [62], does not have a working implementation in gymnax [63].

## 6.2 Single-Task Training Results

Figure 2 shows the performance of OPEN after single-task training. In three MinAtar environments, OPEN **significantly outperform all baselines**, far exceeding previous attempts at learned optimization for RL. Additionally, OPEN beat both learned optimizers in *every* environment. Overall, these experiments show the capability of OPEN to learn highly performant update rules for RL.

Return curves (Appendix E) show that OPEN does not always achieve high returns from the start of training. Instead, OPEN can sacrifice greedy, short-term gains in favor of long-term final return, possibly due to its dynamic update rule. We further analyze the optimizers' behavior in Appendix F.

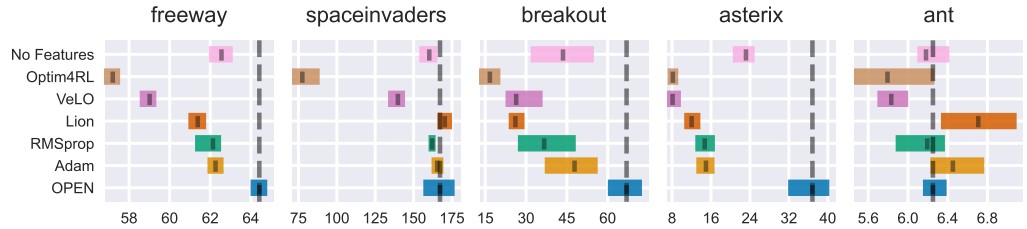

Figure 2: IQM of final returns for the five single-task training environments, evaluated over 16 random environment seeds. We plot 95% stratified bootstrap confidence intervals for each environment.

## 6.3 Multi-Task Training Results

Figure 3 shows each optimizer's ability to fit to multiple MinAtar environments [63, 62], where handcrafted optimizers are tuned *per-environment*. We normalize returns with respect to Adam, and aggregate scores over environments to give a single performance metric. Here, we increase the size of the learned optimizers, with details in Appendix B.5. In addition to IQM, we consider the *mean* normalized return to explore the existence of outliers (which often correspond to asterix, where OPEN strongly outperforms Adam), and optimality gap, a metric from Agarwal et al. [70] measuring how close to optimal algorithms are. Unlike single-task training, where optimizers are trained until convergence, we run multi-task experiments for a fixed number of generations (300) to limit compute.

OPEN drastically outperforms **every** optimizer for multi-task training. In particular, OPEN produces the only optimizer with an aggregate score higher than Adam, demonstrating its ability to learn highly expressive update rules which can fit to a range of contexts; no other learned optimizers get close to OPEN in fitting to multiple environments simultaneously. As expected, OPEN prioritizes optimization in asterix and breakout, according to the return curves (Appendix G); we believe better curricula would help to overcome this issue. Interestingly, Optim4RL seems to perform better in the multi-task setting than the single-task setting. This may be due to the increased number of meta-training samples.

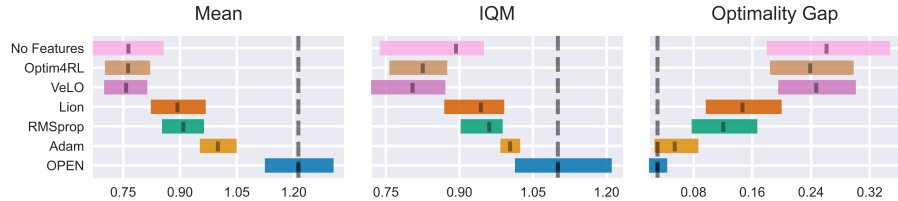

Figure 3: Mean, IQM and optimality gap (smaller = better), evaluated over 16 random seeds *per environment* for the aggregated, Adam-normalized final returns after multi-task training on MinAtar [62, 63]. We plot 95% stratified bootstrap confidence intervals for each metric.

## 6.4 Generalization

Figure 4 shows OPEN's ability to generalize in a gridworld setting. We explore *in-distribution* generalization on the left, and *OOS* generalization on the right, where the top row (*rand_dense*, *rand_sparse* and *rand_long*) are from LPG [13, 66] and the bottom row are 3 mazes from Minigrid [67, 66]. We explore an additional dimension of OOS generalization in the agent's network hidden size; OPEN only saw agents with network hidden sizes of 16 in training, but is tested on larger and smaller agents. We include similar tests for OOS training lengths in Appendix H. We normalize OPEN against Adam, which was tuned for the same distribution and agent that OPEN learned for.

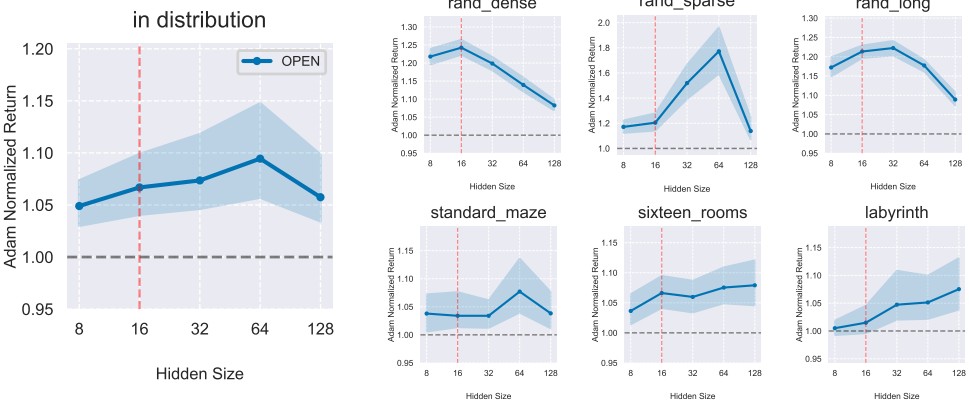

Figure 4: IQM of return, normalized by Adam, in seven gridworlds, with 95% stratified bootstrap confidence intervals for 64 random seeds. On the left, we show performance in the distribution OPEN and Adam were trained and tuned in. On the right, we show OOS performance: the top row shows gridworlds from Oh et al. [13], and the bottom row shows mazes from Chevalier-Boisvert et al. [67]. We mark Hidden Size = 16 as the in-distribution agent size for OPEN and Adam.

OPEN learns update rules that consistently outperform Adam *in-distribution* and *out-of-support* with regards to both the agent and environment; in ***every*** OOS environment and agent size, OPEN outperformed Adam. In fact, in all but rand_dense, its generalization **improves** compared to Adam for some different agent widths, increasing its normalized return. In combination with our findings from Section 6.3, which demonstrate our approach can learn expressive update rules for hard, diverse environments, these results demonstrate the effectiveness of OPEN: if trained on a wide enough distribution, OPEN has the potential to generalize across a wide array of RL problems.

Finally, in Figure 5, we test how the optimizer trained above (i.e., for Figure 4) transfers to Craftax-Classic, an environment with completely different dynamics, *0-shot*. We compare against two baselines: Adam (0-shot), where we tune the hyperparameters of Adam on the distribution of gridworlds that OPEN was trained on; and Adam (Tuned), where the Adam and PPO hyperparameters are tuned directly in Craftax-Classic. The latter gives an idea of the maximum achievable return.

We find that OPEN transfers significantly better than Adam when both are tuned in the same distribution of environments, and performs only marginally worse than Adam tuned directly in-distribution. These results exemplify the generalization capabilities of OPEN and lay the foundation for learning a large scale 'foundation' optimizer which could be used for a wide range of RL problems.

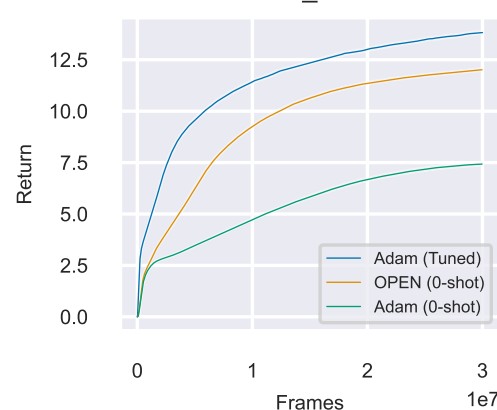

Figure 5: A comparison of OPEN and Adam with and without hyperparameter tuning in Craftax-Classic. We plot mean return over 32 seeds. Standard error is negligible ($< 0.06$).

## 7 Ablation Study

In our ablation study, we explore how each constituent of OPEN contributes to improved performance. Firstly, we focus on two specific, measurable challenges: plasticity loss and exploration. Subsequently, we verify that OPEN can be meta-trained for an RL algorithm besides PPO.

### 7.1 Individual Ablations

We ablate each individual design decision of OPEN in Figure 6. For each ablation, we train 17 optimizers in a shortened version of Breakout [62, 63] and evaluate performance after 64 PPO training runs per optimizer. Further details of the ablation methodology are in Appendix I.

While measuring plasticity loss is important, dormancy alone is not an appropriate performance metric; a newly initialized network has near-zero dormancy but poor performance. Instead, we include dormancy here as one *possible* justification for why some elements of OPEN are useful.

**(P1)** Ablating *training proportion* directly disables the optimizer's ability to tackle *training* non-stationarity. Similarly, removing *batch proportion* prevents dynamic behavior within a (stationary) batch. The corresponding decreases in performance show that both timescales of non-stationarity are beneficial, potentially overcoming the impact of non-stationarity in RL.

**(P2)** Removing *dormancy* as an input has a **drastic** impact on the agent's return, corresponding to a large increase in plasticity loss. While dormancy plays no *direct* role in the optimizer's meta-objective, including it as an input gives the optimizer the *capability* to react as neurons grow dormant, as desired.

**(P2)** Not conditioning on layer proportion has a negative effect on final return. The increase in dormancy from its removal suggests that allowing the optimizer to behave differently for each layer in the network has some positive impact on plasticity loss.

**(P2)/(P3)** Stochasticity benefits are two-fold: besides enhancing exploration (Section 7.2), it notably lowers dormancy. Though limited to the actor, this reduction likely also contributes to improve return.

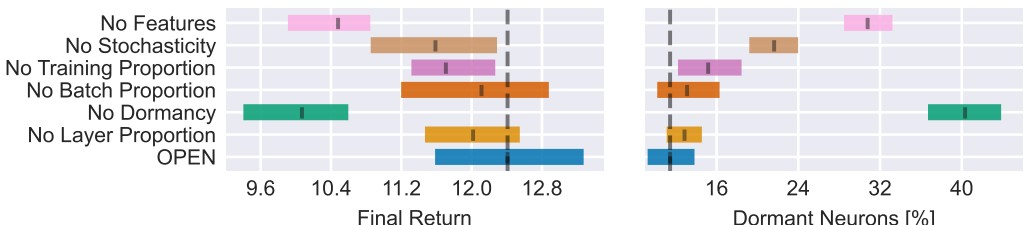

Figure 6: IQM of mean final return for 17 trained optimizers *per ablation*, evaluated on 64 random seeds each, alongside mean $\tau = 0$ dormancy for optimizers in the interquartile range. We show 95% stratified bootstrap confidence intervals.

## 7.2 Exploration

**(P3)** We verify the benefits of *learnable stochasticity* in Deep Sea, an environment from bsuite [71, 63] designed to analyze *exploration*. This environment returns a small penalty for each `right` action but gives a large positive reward if `right` is selected for *every* action. Therefore, agents need to explore beyond the local optimum of 'left at each timestep' to maximize return. Deep Sea's size can be varied, such that an agent needs to take more consecutive `right` actions to receive reward.

Naïvely applying OPEN to Deep Sea struggles; we posit that optimizing *towards* the gradient can be detrimental to the actor since the gradient leads to a local optimum, whereas in the critic the gradient always points to (beneficially) more accurate value functions. Therefore, we augment OPEN to learn different updates for the actor and critic ('*separated*'). In Figure 7, we show the disparity between

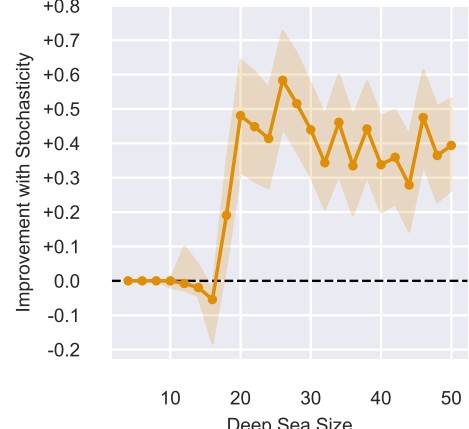

Figure 7: IQM performance improvement of an optimizer with learnable stochasticity over one without. We plot 95% stratified bootstrap confidence intervals over 128 random seeds.

a separated optimizer trained *with* and *without* learnable stochasticity after training for sizes between $[4, 26]$ in a small number of generations (48), noting that larger sizes are OOS for the optimizer. We include full results in Appendix I.2.

The optimizer with learnable stochasticity consistently achieves higher return in large environments compared to without, suggesting significant exploration benefits. In fact, our analysis (Appendix F.5) finds that, despite being unaware of size, stochasticity increases in larger environments. In other words, the optimizer promotes *more* exploration when size increases. However, stochasticity does

marginally reduce performance in smaller environments; this may be due to the optimizer promoting exploration even after achieving maximum reward, rather than converging to the optimal policy.

### 7.3 Alternative RL Algorithm

To ensure that OPEN can learn for algorithms beyond PPO, and thus is applicable to a wider range of scenarios, we explore learning an optimizer for PQN [72] in Figure 8. PQN is a vectorized and simplified alternative to DQN [73]. We include analysis and the return curve in Appendix I, and hyperparameters in Appendix C.

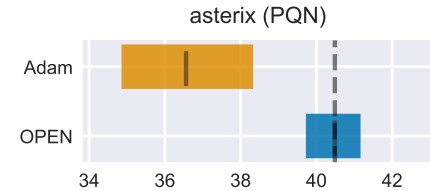

Figure 8: IQM of final return for Adam and OPEN after training PQN in Asterix. We plot 95% stratified bootstrap confidence intervals over 64 seeds.

We find that, in our single meta-training run, OPEN is able to outperform Adam by a significant margin. Therefore, while PPO was the principle RL algorithm for our experiments, our design decisions for OPEN are applicable to a wider class of RL algorithms. While exploring the ability of OPEN to learn for other common RL algorithms would be useful, Figure 8 provides a confirmation of the versatility of our method.

## 8 Limitations and Future Work

While OPEN demonstrates strong success in learning a multi-task objective, our current approach of normalizing returns by Adam biases updates towards environments where Adam underperforms learned optimizers. We believe developing better curricula for learning in this settings, akin to unsupervised environment design [66, 74], would be highly impactful future work. This may unlock the potential for larger scale experiments with wider distributions of meta-training environments. Additionally, the OPEN framework need not be limited to the specific difficulties we focus on here. Exploring ways to include other difficulties of RL which can be measured and incorporated into the framework (e.g., sample efficiency or generalization capabilities of a policy) could potentially elevate the usefulness of OPEN even further. Finally, while we show that OPEN can learn for different RL algorithms, further testing with other popular algorithms (e.g., SAC [75], A2C [76]) would be insightful. Furthermore, training OPEN on many RL algorithms simultaneously could unlock a truly generalist learned optimization algorithm for RL. Successfully synthesizing these proposals with additional scale will potentially produce a universal and performant learned optimizer for RL.

## 9 Conclusion

In this paper, we set out to address some of the major difficulties in RL by meta-learning update rules directly for RL. In particular, we focused on three main challenges: non-stationarity, plasticity loss, and exploration. To do so, we proposed OPEN, a method to train parameterized optimizers that conditions on a set of inputs and uses learnable stochasticity in its output, to specifically target these difficulties. We showed that our method outperforms a range of baselines in four problem settings designed to show the expressibility and generalizability of learned optimizers in RL. Finally, we demonstrated that each design decision of OPEN improved the performance of the optimizer in an ablation study, that the stochasticity in our update expression significantly benefited exploration, and that OPEN was sufficiently versatile to learn with different RL algorithms.

### Author Contributions

**AG** implemented the code, designed and performed all experiments and made the architectural and training design decisions of OPEN. **AG** also provided all analysis and wrote all of the paper.

**CL** proposed the initial project, contributed the initial implementation framework and provided assistance throughout the project's timeline.

**MJ** contributed to experimental and ablation design and helped with editing the manuscript.

**SW** and **JF** provided advice throughout the project, and in writing the paper.

## Acknowledgements

We thank Benjamin Ellis for offering guidance throughout this project, and Pablo Samuel Castro for providing advice in the early stages of the research. We are also grateful to Michael Matthews, Scott le Roux, Juliusz Ziomek and our reviewers for their comments and feedback.

## Funding

**AG** and **MJ** are funded by the EPSRC Centre for Doctoral Training in Autonomous Intelligent Machines and Systems EP/S024050/1. **MJ** is also sponsored by Amazon Web Services. **JF** is partially funded by the UKI grant EP/Y028481/1 (originally selected for funding by the ERC). **JF** is also supported by the JPMC Research Award and the Amazon Research Award. Our experiments were also made possible by a generous equipment grant from NVIDIA.

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

# A  Additional Details: Difficulties in RL

**(Problem 1) Non-stationarity**    Igl et al. [8] highlights different sources of non-stationarity across a range of RL algorithms. For PPO, non-stationary arises due to a changing state visitation distribution, the target value function $V^\pi(s)$ depending on the updating policy and bootstrapping from the generalized advantage estimate [77]. Since PPO batches rollouts, non-stationarity occurs between each batch; within the same batch, the agent solves a *stationary problem*. As the non-stationarity occurs over the course of training, we denote it *training non-stationarity*.

Optimizers designed for stationarity, like Adam [18], have been shown to struggle in non-stationary settings [9] and techniques to deal with non-stationarity have been proposed. Asadi et al. [78] prevent *contamination* between batches by resetting their optimizer state, including resetting their momentum. However, this handcrafted approach is potentially too severe, as it fails to take advantage of *potentially* useful commonality between batches.

**(Problem 2) Plasticity loss**    Plasticity loss, which refers to an inability to fit to new objective over the course of training, is an important problem in RL. As an agent learns, both its input and target distributions change (i.e., nonstationarity, **(P1)**). This means the agent needs to fit new objectives during training, emphasizing the importance of maintaining plasticity *throughout* training. Therefore, there have been many handcrafted attempts to reduce plasticity loss in RL. Abbas et al. [53] find that different activation functions can help prevent plasticity loss, Obando-Ceron et al. [79] suggest using smaller batches to increase plasticity and Nikishin et al. [60] introduces new output heads throughout training. Many have demonstrated the effectiveness of resetting parts of the network [80, 59, 81, 49], though this runs the risk of losing *some* previously learned, useful information. Additionally, these approaches are all *hyperparameter-dependent* and unlikely to eliminate plasticity loss robustly.

**(Problem 3) Exploration**    Exploration in RL has a rich history of methodologies: [82–84], to name a few. While there are simple, heuristic approaches, like $\epsilon$-greedy [1], recent methods have been designed to deal with complex, high dimensional domains. These include count based methods [85, 86], learned exploration models [87, 88] or using variational techniques [89]. Parameter space noise [11] involves noising each parameter in an agent to enable exploration across different rollouts while behaving consistently within a given rollout. This inspired our approach to exploration since it is algorithm-agnostic and can be implemented directly in the optimizer.

# B Optimizer Details

## B.1 Algorithm

We detail an example update step when using OPEN. We use notation from Appendix B.2, and use $z_t^{GRU}$ to refer to the GRU hidden state at update $t$. In this algorithm, we assume the dormancy for each neuron has been *tiled* over the relevant parameters. In practice, we build our optimizer around the library fromMetz et al. [45].

---

**Algorithm 1:** Example update step from OPEN.

---

**Given:** opt, $N_{batch}$, $N_{minibatches}$, $L$, $H$, $\beta$, $\alpha_1$, $\alpha_2$, $\alpha_3$
**Input:** $\mathbf{g}_t$, $\mathbf{m}_{t-1}$, $\mathbf{p}_t$, $t$, $\mathbf{dormancies}_t$, $\mathbf{h}$, $\mathbf{z}_{t-1}^{GRU}$
**Output:** $(\mathbf{p}_{t+1}, \mathbf{m_t}, \mathbf{z}_t^{GRU})$

/* Compute new momentums                                                                */
**for** *each $\beta$* **do**
  $\quad m_t^\beta = \beta \times m_{t-1}^\beta + (1 - \beta) \times g_t$
**end**
/* Compute training and batch proportions                                              */
$TP = (t//(L * N_{minibatches})/N_{batch}$
$BP = ((t//N_{minibatches}) \bmod L)/L$
/* Compute first stage of update                                                       */
**for** *each parameter $i$ in agent* **do**
  $\quad$ input = $[\text{sgn}(g_{(t,i)}), \log(g_{(t,i)}), \text{sgn}(\boldsymbol{m}_{(t,i)}), \log(\boldsymbol{m}_{(t,i)}), p_{(t,i)}, TP, BP, D_{(t,i)}, h_i]$
  $\quad$ GRU$_{out}$, $z_{(t,i)}^{GRU}$ = opt$^{GRU}$(input, $z_{(t-1,i)}^{GRU}$)
  $\quad m_i, e_i, \delta_i = \text{opt}^{MLP}(\text{GRU}_{out})$
  $\quad \hat{u}_i = \alpha_1 m_i \exp \alpha_2 e_i$
  $\quad$ **if** *$i$ in actor* **then**
    $\quad\quad \hat{u}_i = \hat{u}_i + \alpha_3 \delta_i \epsilon$
  $\quad$ **end**
**end**
/* Zero-mean updates and apply them                                                    */
**for** *each parameter $i$ in agent* **do**
  $\quad u_i = \hat{u}_i - \mathbb{E}_i[\hat{u}_i]$
  $\quad p_{(t+1,i)} = p_{(t,i)} - u_i$
**end**
**return** $(\mathbf{p}_{t+1}, \mathbf{m_t}, \mathbf{z}_t^{GRU})$

---

## B.2 Input Features to the Optimizer

Our full list of features is concatenated and fed into the optimizer, as in algorithm 1 below. We use some notation from Table 3. These features are:

- Gradient $g$, processed with the transformation $g \to \{sign(g), \log(g + \epsilon)\}$.

- Momentum $m$ calculated with the following $\beta$ coefficients: $[0.1, 0.5, 0.9, 0.99, 0.999, 0.9999]$. This is processed with the transformation $m_i \to \{sign(m_i), \log(m_i + \epsilon)\}$ for each $\beta_i$.

- Current value of the parameter $p$.

- Training proportion, or how far the current batch is through the total number of batches. Since the total number of batches is calculated as $N_{batch} = (T//N_{steps})//N_{envs}$, this is defined as $(t//(L * N_{minibatches})/N_{batch}$, where $t$ is the current update iteration. We use a floor division operator as training proportion should be constant through a batch.

- Batch proportion, or how far the current update is through training with the same batch, calculated as $((t//N_{minibatches}) \bmod L)/L$.

- Dormancy, $D$, calculated for the neuron downstream of the parameter using equation 6. This is repeated over parameters with the same downstream neuron.
- Proportion of the current layer through the network, $h/H$.

The gradient, momentum and parameter value features are normalized to have a second moment of 1 across the tensor, following Metz et al. [90, 45], as a normalization strategy which preserves *direction*. Like Lan et al. [17], we process gradients and momentum as $x \rightarrow \{\text{sign}(x), \log(x + \epsilon)\}$ to magnify the difference between small values.

## B.3   Optimizer Architecture Figure

Figure 9 visually demonstrates the architecture of OPEN . Since only the actor is updated with learned stochasticity ($\delta^{\text{actor}}$), we multiply the sampled noise $\epsilon$ with a $1/0$ mask for the actor/critic.

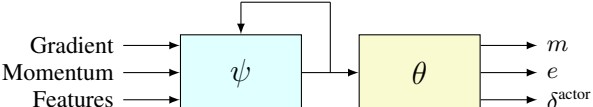

Figure 9: A visualization of the architecture used by OPEN. $\psi$ are parameters of a GRU and $\theta$ are parameters of an MLP. Since $\delta^{\text{actor}}$ only applies to the actor's update expression, we set the noise to zero for the critic to keep its updates deterministic.

## B.4   Single-Task and Gridworld Optimizer Architecture Details

The details of the layer sizes of OPEN in both the single-task and gridworld settings are in Table 1.

Table 1: Optimizer layer sizes for OPEN in the single-task and gridworld experiments.

| Layer Type | Dimensionality |
|---|---|
| *GRU* | $[19, 8]$ |
| *Fully* Connected | $[8, 16]$ |
| *Layernorm* | – |
| *Fully Connected* | $[16, 16]$ |
| *Layernorm* | – |
| *Fully Connected* | $[16, 3]$ |

In our update rule, we let $\alpha_1 = \alpha_2 = \alpha_3 = 0.001$. We mask out the third output in the critic by setting the noise ($\epsilon$) to zero; otherwise, updates to our critic would be non-deterministic.

We use layers of the same size in '*No Features*', but condition on less inputs (14) and do not have the third output. We use the same architecture as Lan et al. [17] for Optim4RL, which includes two networks, each comprising a size 8 GRU, and two size 16 fully connected layers.

## B.5   Multi-Task Optimizer Architecture Details

When training optimizers for the multi-task setting (Section 6.3), we find increasing the network sizes consistently benefits performance. We show the architecture for OPEN in this experiment in Table 2, masking out the noise for the critic by setting $\epsilon = 0$. We also enlarge *No Features* and Optim4RL by doubling the GRU sizes to 16 and the hidden layers to size 32.

Table 2: Optimizer layer sizes for OPEN in the multi-task experiment.

| Layer Type | Dimensionality |
|---|---|
| *GRU* | $[19, 16]$ |
| *Fully* Connected | $[16, 32]$ |
| *Layernorm* | – |
| *Fully Connected* | $[32, 32]$ |
| *Layernorm* | – |
| *Fully Connected* | $[32, 3]$ |

# C  Hyperparameters

## C.1  PPO Hyperparameters

PPO hyperparameters for the environments included in our experiments are shown in Table 3. For our gridworld experiments, we learned OPEN and tuned Adam [18] for a PPO agent with $W = 16$, but evaluated on widths $W \in [8, 16, 32, 64, 128]$. Hyperparameters for PPO are taken from [32] where possible.

Table 3: PPO hyperparameters. All MinAtar environments used common PPO parameters, and are thus under one header.

| Hyperparameter | Environment | | |
|---|---|---|---|
| | **MinAtar** | **Ant** | **Gridworld** |
| *Number of Environments $N_{envs}$* | 64 | 2048 | 1024 |
| *Number of Environment Steps $N_{steps}$* | 128 | 10 | 20 |
| *Total Timesteps $T$* | $1 \times 10^7$ | $5 \times 10^7$ | $3 \times 10^7$ |
| *Number of Minibatches $N_{minibatch}$* | 8 | 32 | 16 |
| *Number of Epochs $L$* | 4 | 4 | 2 |
| *Discount Factor $\gamma$* | 0.99 | 0.99 | 0.99 |
| *GAE $\lambda$* | 0.95 | 0.95 | 0.95 |
| *PPO Clip $\epsilon$* | 0.2 | 0.2 | 0.2 |
| *Value Function Coefficient $c_1$* | 0.5 | 0.5 | 0.5 |
| *Entropy Coefficient $c_2$* | 0.01 | 0 | 0.01 |
| *Max Gradient Norm* | 0.5 | 0.5 | 0.5 |
| *Layer Width $W$* | 64 | 64 | 16 |
| *Number of Hidden Layers $H$* | 2 | 2 | 2 |
| *Activation* | ReLU | tanh | tanh |

## C.2  Optimization Hyperparameters

For Adam, RMSprop and Lion, we tune hyperparameters with fixed PPO hyperparameters. For each environment we run a sweep and evaluate performance over four seeds (eight in the gridworld), picking the combination with the highest final return. In many cases, we found the optimizers to be fairly robust to reasonable hyperparameters. For Lion [20], we search over a learning rate range from 3-10× smaller than Adam and RMSprop as suggested by Chen et al. [20]. For Optim4RL [17], hyperparameter tuning is too expensive; instead, we set learning rate to be $0.1 \times \text{LR}_{\text{Adam}}$ following the heuristic from [20].

For Adam, RMSprop and Lion, we also test whether annealing learning rate from its initial value to 0 over the course of training would improve performance.

A full breakdown of which hyperparameters are tuned, and their value, is shown in the following tables. We use notation and implementations for each optimizer from optax [69].

Table 4: Optimization hyperparameters for MinAtar environments. '**Range**' covers the range of values used in our hyperparameter tuning.

| Optimizer | Hyper Parameter | Environment | | | | Range |
|---|---|---|---|---|---|---|
| | | asterix | freeway | breakout | space invaders | |
| **Adam** | $LR$ | 0.003 | 0.001 | 0.01 | 0.007 | $\{0.001, 0.01\}$ |
| | $\beta_1$ | 0.9 | 0.9 | 0.9 | 0.9 | $\{0.9, 0.999\}$ |
| | $\beta_2$ | 0.999 | 0.99 | 0.99 | 0.99 | $\{0.9, 0.999\}$ |
| | *Anneal LR* | ✓ | ✓ | ✓ | ✓ | $\{✓, ✗\}$ |
| **RMSprop** | $LR$ | 0.002 | 0.001 | 0.002 | 0.009 | $\{0.001, 0.01\}$ |
| | *Decay* | 0.99 | 0.999 | 0.99 | 0.99 | $\{0.9, 0.999\}$ |
| | *Anneal LR* | ✓ | ✓ | ✗ | ✓ | $\{✓, ✗\}$ |
| **Lion** | $LR$ | 0.0003 | 0.0003 | 0.0008 | 0.0008 | $\{0.0001, 0.001\}$ |
| | $\beta_1$ | 0.99 | 0.9 | 0.9 | 0.9 | $\{0.9, 0.999\}$ |
| | $\beta_2$ | 0.9 | 0.9 | 0.9 | 0.99 | $\{0.9, 0.999\}$ |
| | *Anneal LR* | ✓ | ✓ | ✗ | ✓ | $\{✓, ✗\}$ |
| **Optim4RL** | $LR$ | 0.0003 | 0.0001 | 0.001 | 0.0007 | $\{0.1 \times \text{LR}_{\text{Adam}}\}$ |

Table 5: Optimization hyperparameters for Ant. '**Range**' covers the range of values used in our hyperparameter tuning.

| Optimizer | Hyperparameter | Environment | Range |
|---|---|---|---|
| | | Ant | |
| **Adam** | $LR$ | 0.0003 | $\{0.0001, 0.001\}$ |
| | $\beta_1$ | 0.99 | $\{0.9, 0.999\}$ |
| | $\beta_2$ | 0.99 | $\{0.9, 0.999\}$ |
| | *Anneal LR* | ✓ | $\{✓, ✗\}$ |
| **RMSprop** | $LR$ | 0.0008 | $\{0.0001, 0.005\}$ |
| | *Decay* | 0.99 | $\{0.9, 0.999\}$ |
| | *Anneal LR* | ✗ | $\{✓, ✗\}$ |
| **Lion** | $LR$ | 0.00015 | $\{0.00001, 0.0005\}$ |
| | $\beta_1$ | 0.9 | $\{0.9, 0.999\}$ |
| | $\beta_2$ | 0.9 | $\{0.9, 0.999\}$ |
| | *Anneal LR* | ✓ | $\{✓, ✗\}$ |
| **Optim4RL** | $LR$ | 0.00003 | $\{0.1 \times \text{LR}_{\text{Adam}}\}$ |

Table 6: Optimization hyperparameters for Gridworld. '**Range**' covers the range of values used in our hyperparameter tuning.

| Optimizer | Hyperparameter | Environment | Range |
|---|---|---|---|
| | | Gridworld | |
| **Adam** | $LR$ | 0.0001 | $\{0.00005, 0.001\}$ |
| | $\beta_1$ | 0.99 | $\{0.9, 0.999\}$ |
| | $\beta_2$ | 0.99 | $\{0.9, 0.999\}$ |
| | *Anneal LR* | ✓ | $\{✓, ✗\}$ |

### C.3 Craftax-Classic Hyperparameters

For Craftax-Classic (Table C.3), we test 0-shot optimizers using the gridworld hyperparameters to ensure transferring between the settings is as fair as possible. For the finetuned case, we use PPO hyperparameters from [21] which were tuned in domain. Also, for the finetuned case we set $\beta_1$ and $\beta_2$ values to defaults in optax [69] and tune the learning rate in the range $\{0.00005, 0.0005\}$. All optimizers are set to $W = 64$ so as to keep the network size close to, but still out of, distribution.

Table 7: Hyperparameters for Craftax-Classic.

| Hyperparameter | OPEN (0-shot) | Adam (0-shot) | Adam (Finetuned) |
|---|---|---|---|
| *Learning rate* | — | 0.0001 | 0.0005 |
| $\beta_1$ | — | 0.99 | 0.9 |
| $\beta_2$ | — | 0.99 | 0.999 |
| *Anneal LR* | — | ✓ | ✓ |
| *Number of Environments* $N_{envs}$ | 1024 | 1024 | 256 |
| *Number of Environment Steps* $N_{steps}$ | 20 | 20 | 16 |
| *Total Timesteps* $T$ | $3 \times 10^7$ | $3 \times 10^7$ | $3 \times 10^7$ |
| *Number of Minibatches* $N_{minibatch}$ | 16 | 16 | 8 |
| *Number of Epochs* $L$ | 2 | 2 | 4 |
| *Discount Factor* $\gamma$ | 0.99 | 0.99 | 0.99 |
| *GAE* $\lambda$ | 0.8 | 0.8 | 0.8 |
| *PPO Clip* $\epsilon$ | 0.2 | 0.2 | 0.2 |
| *Value Function Coefficient* $c_1$ | 0.5 | 0.5 | 0.5 |
| *Entropy Coefficient* $c_2$ | 0.01 | 0.01 | 0.01 |
| *Max Gradient Norm* | 0.5 | 0.5 | 0.5 |
| *Layer Width* $W$ | 64 | 64 | 64 |
| *Number of Hidden Layers* $H$ | 2 | 2 | 2 |
| *Activation* | tanh | tanh | tanh |

### C.4 PQN Hyperparameters

For PQN, we use algorithm hyperparameters from [72]. For experiments with Adam, we tune learning rate, $\beta_1$ and $\beta_2$ and *Anneal LR*.

Table 8: Hyperparameters for PQN in asterix.

| Hyperparameter | OPEN | Adam | Tuning Range |
|---|---|---|---|
| *Learning rate* | — | 0.0003 | $\{0.0001, 0.001\}$ |
| $\beta_1$ | — | 0.99 | $\{0.9, 0.999\}$ |
| $\beta_2$ | — | 0.999 | $\{0.9, 0.999\}$ |
| *Anneal LR* | — | ✓ | $\{✓, ✗\}$ |
| *Number of Environments* $N_{envs}$ | 128 | 128 | |
| *Number of Environment Steps* $N_{steps}$ | 32 | 32 | |
| *Total Timesteps* $T$ | $1 \times 10^7$ | $1 \times 10^7$ | |
| *Number of Minibatches* $N_{minibatch}$ | 32 | 32 | |
| *Number of Epochs* $L$ | 4 | 4 | |
| *Discount Factor* $\gamma$ | 0.99 | 0.99 | |
| *GAE* $\lambda$ | 0.65 | 0.65 | |
| *Max Gradient Norm* | 10 | 10 | |
| *Layer Width* $W$ | 128 | 128 | |
| *Number of Hidden Layers* $H$ | 2 | 2 | |
| *Activation* | relu | relu | |
| *Normalization* | LayerNorm | LayerNorm | |

### C.5 ES Hyperparameters

Due to the length of meta-optimization, it is not practical to tune hyperparameters for meta-training. We, however, find the following hyperparameters effective and robust for our experiments. We use common hyperparameters when learning OPEN, Optim4RL and *No Features*.

For the single-task and gridworld settings, we train all optimizers for a number of generations after their performance stabilizes, which took different times between optimizers, to ensure each learned optimizer has *reached convergence*; this occasionally means optimizers were trained for different number of generations, but enables efficient use of computational resources by not continuing training far past any performance gains can be realized. For the multi-task setting, due to the extreme cost of training (i.e., having to evaluate sequentially on four MinAtar environments), we train each optimizer

with an equivalent level of compute (i.e., 300 generations) and pick the best performing generation in that period.

We periodically evaluate the learned optimizers during training to find the one which generation performs best on a small *in-distribution* validation set. Table 9 shows roughly how many generations we *trained* each optimizer for, and the generation used at test time, which varied for each environment. We find that in some environments, Optim4RL made no learning progress from the beginning of training, with its performance being practically identical to its initialization.

Table 9: ES hyperparameters for meta-training. For MinAtar, the number of generations corresponds to {Asterix, Breakout, Freeway, SpaceInvaders}. We show the *max generations* for each optimizer (i.e., how long it was trained for), as well as the *generation used* (i.e., which training generation was used at inference time).

| Hyperparameter | Environments | | | |
| --- | --- | --- | --- | --- |
| | MinAtar | Ant | Multi-Task | Gridworld |
| $\sigma_{init}$ | 0.03 | 0.01 | 0.03 | 0.01 |
| $\sigma_{decay}$ | 0.999 | 0.999 | 0.998 | 0.999 |
| *Learning Rate* | 0.03 | 0.01 | 0.03 | 0.005 |
| *Learning Rate Decay* | 0.999 | 0.999 | 0.998 | 0.990 |
| *Population Size* | 64 | 32 | 64 | 64 |
| *Number of Rollouts* | 1 | 1 | 1 | 1 |
| *Max Gens* (OPEN) | $\sim \{350, 500, 700, 450\}$ | $\sim 700$ | 300 | $\sim 350$ |
| *Gen Used* (OPEN) | $\{336, 312, 648, 144\}$ | 168 | 234 | 333 |
| *Max Gens (Optim4RL)* | $\sim \{500, 550, 300, 400\}$ | $\sim 700$ | 300 | N/A |
| *Gen Used (Optim4RL)* | $\{288, 552, 216, 24\}$ | 480 | 288 | N/A |
| *Max Gens (No Features)* | $\sim \{800, 850, 500, 600\}$ | $\sim 400$ | 300 | N/A |
| *Gen Used (No Features)* | $\{456, 816, 408, 264\}$ | 240 | 270 | N/A |
| *Evaluation Frequency* | 24 | 24 | 9 | 9 |

# D Gridworld Details

We train OPEN on gridworlds by sampling environment parameters from a *distribution*. Here, we use a distribution implemented by [66, 32] which is detailed in Table 10. In the codebase of Jackson et al. [66], this is referred to as 'all'.

Table 10: Distribution parameters for Gridworld. In training, the values for the experienced environment are *sampled* from the value range.

| Parameter | Value [range] |
|---|---|
| *Max Steps in Episode* | $[20, 750]$ |
| *Object Rewards* | $[-1.0, 1.0]$ |
| *Object $p(terminate)$* | $[0.01, 1.0]$ |
| *Object $p(respawn)$* | $[0.001, 0.1]$ |
| *Number of Objects* | $[1, 6]$ |
| *Grid Size* | $[4, 11]$ |
| *Number of Walls* | $15$ |

We also evaluate OPEN on three distributions from Oh et al. [13], and three mazes from Chevalier-Boisvert et al. [67], which were not in the distribution of 'all'. We run all experiments on 64 seeds per gridworld. These are all detailed below, with the mazes also visualized.

Table 11: Distribution parameters for Gridworld. Curly brackets denote a list of the true values, corresponding to each object, used in testing.

| Name | Parameter | Values |
|---|---|---|
| **rand_dense** | *Max Steps in Episode* | 500 |
| | *Object Rewards* | $(1.0, 1.0, -1.0, -1.0)$ |
| | *Object $p(terminate)$* | $(0.0, 0.0, 0.5, 0.0)$ |
| | *Object $p(respawn)$* | $(0.05, 0.05, 0.1, 0.5)$ |
| | *Number of Objects* | 4 |
| | *Grid Size* | 11 |
| | *Number of Walls* | 0 |
| **rand_sparse** | *Max Steps in Episode* | 50 |
| | *Object Rewards* | $(-1.0, 1.0)$ |
| | *Object $p(terminate)$* | $(1.0, 1.0)$ |
| | *Object $p(respawn)$* | $(0.0, 0.0)$ |
| | *Number of Objects* | 2 |
| | *Grid Size* | 13 |
| | *Number of Walls* | 0 |
| **rand_long** | *Max Steps in Episode* | 1000 |
| | *Object Rewards* | $(1.0, 1.0, -1.0, -1.0)$ |
| | *Object $p(terminate)$* | $(0.0, 0.0, 0.5, 0.5)$ |
| | *Object $p(respawn)$* | $(0.01, 0.01, 1.0, 1.0)$ |
| | *Number of Objects* | 4 |
| | *Grid Size* | 11 |
| | *Number of Walls* | 0 |

Table 12: Distribution parameters for Gridworld. At inference, true values are *sampled* from the given range, such that each seed is evaluated in a slightly different setting.

| Name | Parameter | Value |
|---|---|---|
| | *Max Steps in Episode* | $[25, 50]$ |
| | *Object Rewards* | $[0.0, 1.0]$ |
| | *Object $p(terminate)$* | $[0.01, 1.0]$ |
| **standard_maze** | *Object $p(respawn)$* | $[0.001, 0.1]$ |
| | *Number of Objects* | 3 |
| | *Grid Size* | 13 |
| | *Max Steps in Episode* | $[25, 50]$ |
| | *Object Rewards* | $[0.0, 1.0]$ |
| | *Object $p(terminate)$* | $[0.01, 1.0]$ |
| **sixteen_rooms** | *Object $p(respawn)$* | $[0.001, 0.1]$ |
| | *Number of Objects* | 3 |
| | *Grid Size* | 13 |
| | *Max Steps in Episode* | $[25, 50]$ |
| | *Object Rewards* | $[0.0, 1.0]$ |
| | *Object $p(terminate)$* | $[0.01, 1.0]$ |
| **labyrinth** | *Object $p(respawn)$* | $[0.001, 0.1]$ |
| | *Number of Objects* | 3 |
| | *Grid Size* | 13 |

We visualize the three mazes in Figure 10.

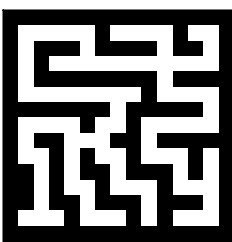 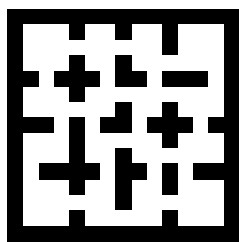 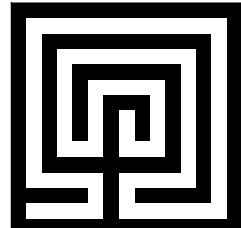

Figure 10: The three mazes from minigrid [67] used for our OOS tests. From left to right, the mazes are: 'standard_maze', 'sixteen_rooms' and 'labyrinth'.

# E Single Environment Return Curves

In Figure 11, which shows return curves over training for the five 'single-task' environments (section 6), OPEN significantly beats 3 of the 5 baselines, performs similarly to Lion [20] in space invaders and performs comparably to hand-crafted optimizers and 'No Features' in ant. OPEN is clearly able to learn strongly performing update rules in a range of single-task settings.

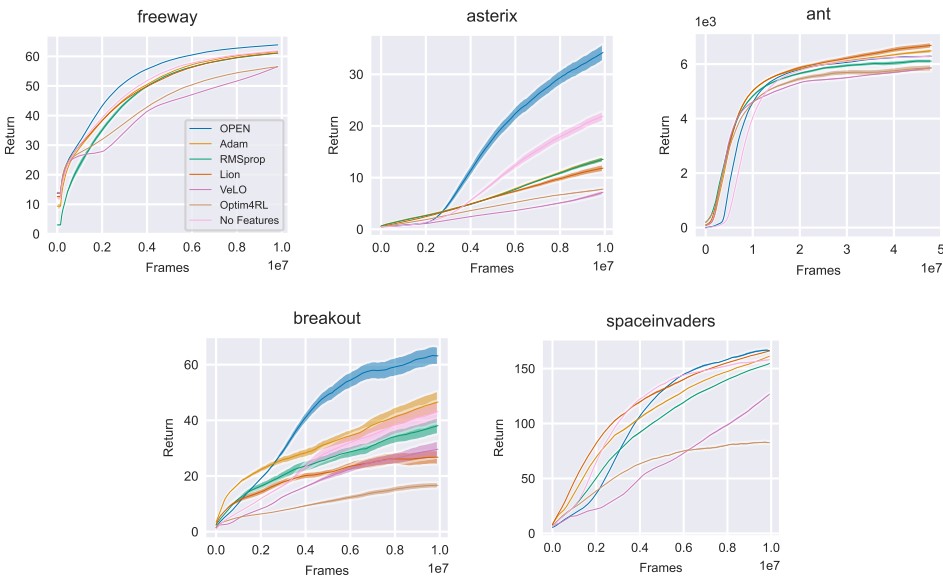

Figure 11: RL training curves comparing our learned optimizers and all other baselines, each trained or tuned on a single environment and evaluated on the same environment. We show mean return over training with standard error, evaluated over 16 seeds.

# F Analysis

Throughout this section, we include analysis and figures exploring the behavior of optimizers learned by OPEN. In particular, we consider the behavior of OPEN with regards to dormancy, momentum, update size and stochasticity in different environments. Where relevant, we also make comparisons to the '*No Features*' optimizer, introduced as a baseline in Section 6, to explore possible differences introduced by the additional elements of OPEN.

Due to the number of moving parts in OPEN, it is difficult to make strong claims regarding the behaviour of the learned optimizers it produces. Instead, we attempt to draw some conclusions based on what we believe the data plots included below suggest, while recognizing that the black-box nature of its update rules, which leads to a lack of interpretability, is a potential drawback of the method in the context of analysis.

All plots included below pertain to the *single task* regime, besides section F.5 which focuses specifically on deep sea.

## F.1 Dormancy

Figure 12 shows the $\tau = 0$ dormancy curves during training for each of the MinAtar [62, 63] environments. These environments were selected as the only ones where the agent uses ReLU activation functions; ant [65, 64] used a tanh activation function for which $\tau = 0$ dormancy is not applicable.

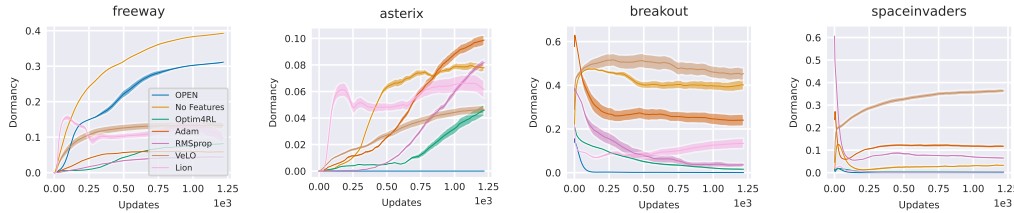

Figure 12: $\tau = 0$ dormancy curves for all optimizers on the four MinAtar environments, evaluated over 16 seeds each. We plot the mean dormancy with standard error.

In all environments excluding freeway, dormancy of *all* agents quickly converged to 0. Additionally, in *every* environment, OPEN produced more plastic (i.e., less dormant) agents than the featureless optimizer. We draw conclusions about this behavior below:

- OPEN optimizes towards *zero* dormancy in a subset of environments, demonstrating it is capable of doing so. The fact that this only held in three out of four environments, despite OPEN outperforming or matching baselines in *every* environment, suggests it optimizes towards zero dormancy only when the agent plasticity is *limiting* performance (i.e., when plasticity loss prevents the agent from improving). Therefore, though a dormancy regularization term in the fitness function for meta-training may have been beneficial for asterix, breakout and spaceinvaders, it is possible that this could harm the performance in freeway and would require expensive hyperparameter tuning of the regularization coefficient to maximize performance. Our meta-learning approach completely sidesteps this problem.

- Despite having the same optimization objective, *No Features* always had higher dormancy than OPEN. In tandem with our ablation from Figure 6, it is likely that the inclusion of each additional feature has given OPEN the capability to minimize plasticity loss.

## F.2 Similarity Between Update and Gradient/Momentum

We evaluate the *cosine similarity* between the update from OPEN and *'No Features'* with momentum over different timescales, and the gradient, in Figure 13. Momentum is calculated as $m^{t+1} = \beta m^t + (1 - \beta)g^t$, where $g_t$ is the gradient at time $t$. In OPEN, inspired by [24], we input momentum using $\beta$s in $[0.1, 0.5, 0.9, 0.99, 0.999, 0.9999]$. For the $i$th timescale, we denote the momentum as $m_i$ in Figure 13. For the gradient, we include an additional comparison against Adam.

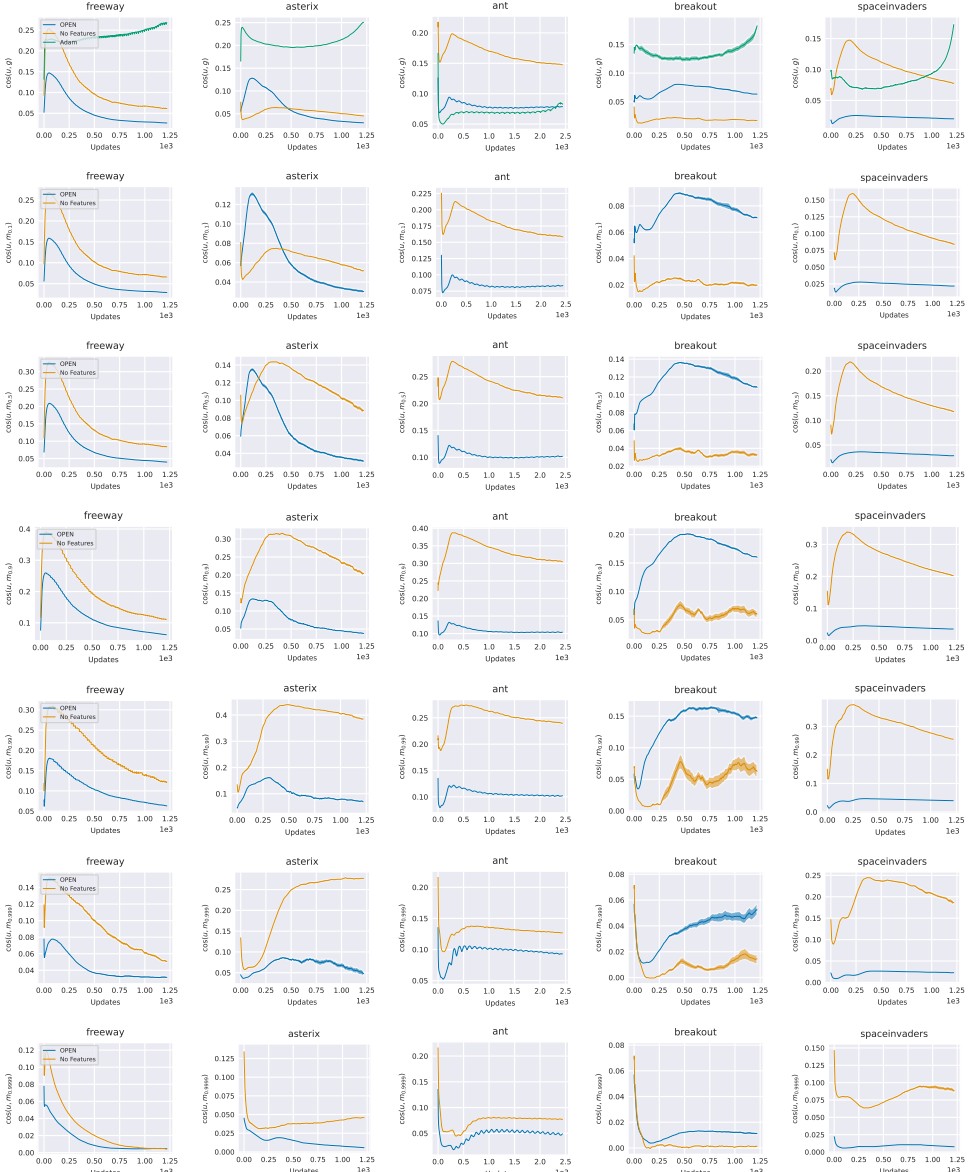

Figure 13: Curves showing the cosine similarity between the updates from OPEN and *'No Features'* with the gradient or momentum at different $\beta$ timescales. Each column corresponds to a different environment, and each row to a different timescale. In order, the rows show cosine similarity with gradient, followed by momentum at $\beta = [0.1, 0.5, 0.9, 0.99, 0.999, 0.9999]$. We plot mean cosine similarities with standard error over 16 runs.

From Figure 13, we can conclude that:

- For both OPEN and '*No Features*', alignment with momentum seems to be maximized at similar (though not always the *same*) timescales as the $\beta_1$ values tuned for Adam in Section C.2.

- OPEN and '*No Features*' rarely had similar cosine similarities. While it is difficult to discern whether this arises as a result of randomness (optimizers trained from a single seed) or something more fundamental, it may demonstrate how the additional elements of OPEN have changed its optimization trajectory.

- For both learned optimizers, cosine similarity with gradient and momentum *generally* peaked and decreased over the training horizon. While OPEN includes lifetime conditioning, which may partially influence this behavior, it is likely that both optimizers rely on their own hidden states, which can be thought of as a learned momentum, more as training progresses.

### F.3 Non-Stochastic Update

In the following experiments (i.e., Appendices F.3, F.4, F.5), we find it useful to divide updates by their respective parameter value. Unlike most handcrafted optimizers which produce shrinking updates from an annealed learned rate and generally output parameters with magnitudes close to zero, OPEN generally increases the magnitude of parameters over training with roughly constant update magnitudes. As such, normalizing updates with respect to the parameter value shows the relative effect (i.e., change) when an update is applied. In all cases, we plot the average absolute normalized value over time.

In Figure 14, we explore how the magnitude of the non-stochastic update changes over time (i.e., the update before any learned stochasticity is applied, defined as $\hat{u}_i = \alpha_1 m_i \exp \alpha_2 e_i$). This is normalized with respect to the parameter value, and so is calculated $|\hat{u}/p| = \mathbb{E}_i\left[\left|\frac{\hat{u}_i}{p_i}\right|\right]$.

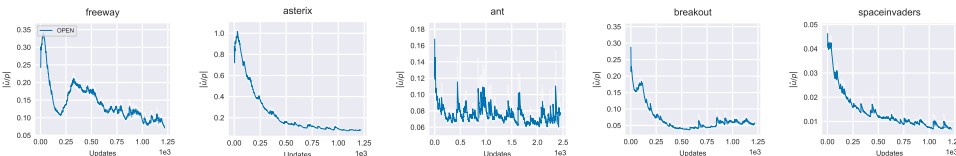

Figure 14: Plots of the average normalized, non-stochastic update magnitude with respect to time, with standard error over 16 seeds.

We find that optimizers learned with OPEN all anneal their step sizes. In particular, the relative size of the update with respect to the parameter size decreases over time, with final updates generally being much smaller than at the beginning of training. This holds in all environments, though was weaker in ant which had significantly smaller updates from the start. Interestingly, for all handcrafted optimizers, hyperparameter tuning also consistently produces smaller learning rates for ant.

### F.4 Stochasticity

Figure 15 explores how the weight of the learned stochasticity, $\delta_i^{\text{actor}}$, changes with respect to time. This stochasticity is incorporated into the update rules as $\hat{u}_i^{\text{actor}} = \hat{u}_i^{\text{actor}} + \alpha_3 \delta_i^{\text{actor}} \epsilon$, with $\epsilon \sim \mathcal{N}(0, 1)$. As in the previous plot, this is normalized with respect to the parameter value, and is calculated as $\text{randomness}/p = \mathbb{E}_i\left[\left|\frac{\delta_i^{\text{actor}}}{p_i^{\text{actor}}}\right|\right]$

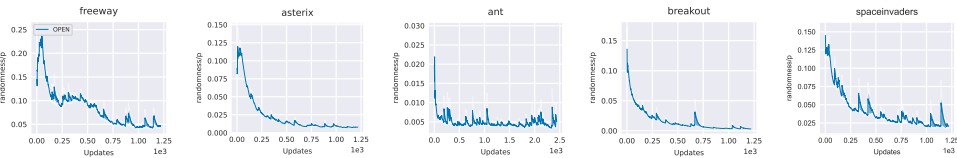

Figure 15: Plots of the normalized stochasticity weight with respect to time. We plot mean values with standard error over 16 seeds.

As expected, this decreases significantly over time, suggesting the optimizers promotes less exploration over time. This makes sense; it is beneficial to explore early in training, when there is time to continue searching for better local or global optima, but later in training it is important to converge to the nearest optimum.

We also note the significantly different scales of noise which are applied to different environments. In particular, ant uses very small noise, suggesting it is not an environment which needs a significant amount of exploration. This may be one reason why OPEN performed similarly to handcrafted optimizers, and '*No Features*'.

### F.5 Deepsea

Finally, we consider how the randomness used by OPEN changes during training in environments of different sizes from Deep sea [71, 63]. This randomness was shown to be crucial for learning in larger environments in Section 7.2. To analyze this, we consider 5 different sizes $(10, 20, 30, 40, 50)$, with the size corresponding to how many 'rights' the agent has to take in a row to receive a large reward. We look at randomness$/p = \mathbb{E}_i \left[ |\frac{\delta_i^{actor}}{p_i^{actor}}| \right]$, as in Appendix F.4.

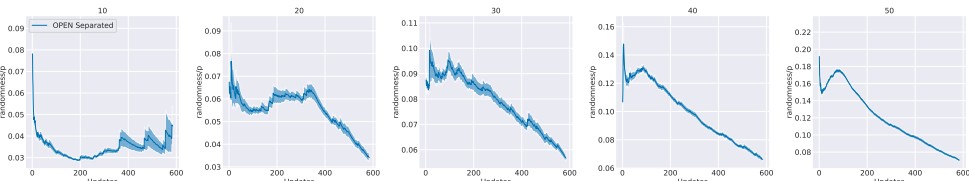

Figure 16: Mean normalized randomness applied by OPEN for different sized Deep Sea environments, over 128 seeds, with standard error. Notably, the scale for larger environments is bigger, such that OPEN applies *more* noise (i.e., more exploration) in larger environments despite having no awareness of the environment size.

We focus on two behaviors of the optimizer. Firstly, the applied randomness progressively decreases over training, reducing the exploration as the agent converges. This follows similar behavior to figure 15, where the level of randomness decreases during the training horizon. Secondly, the amount of noise seems to grow with the environment size; the optimizer promotes more exploration in larger environments, despite lacking any awareness about the environment size. The combination of these two elements allows the agent to *explore* sufficiently in larger environments, while converging to good, close to optimal policies towards the end of training.

# G Multi Environment Return Curves

Figure 17 shows return curves the four MinAtar [62] environments included in 'multi-task' training (Section 6. As expected, the method of normalization used in training (i.e., dividing scores in each environment by scores achieved by Adam and averaging across environments) leads OPEN, and the other learned optimizers, to principally focus on those which they can strongly outperform Adam in. In particular, OPEN marginally underperforms Adam and other handcrafted optimizers in freeway and spaceinvaders. However, its score in asterix, which is approximately double Adam's, will outweigh these other environments. We leave finding better methods for multi-task training, such as using more principled curricula to normalize between environments, to future work. However, we still note that this demonstrates that OPEN is capable of learning highly expressive update rules which can fit to many different contexts; it performs comparatively to handcrafted optimizers in two environments and significantly outperforms them in two others, in addition to consistently matching or beating the other learned baselines.

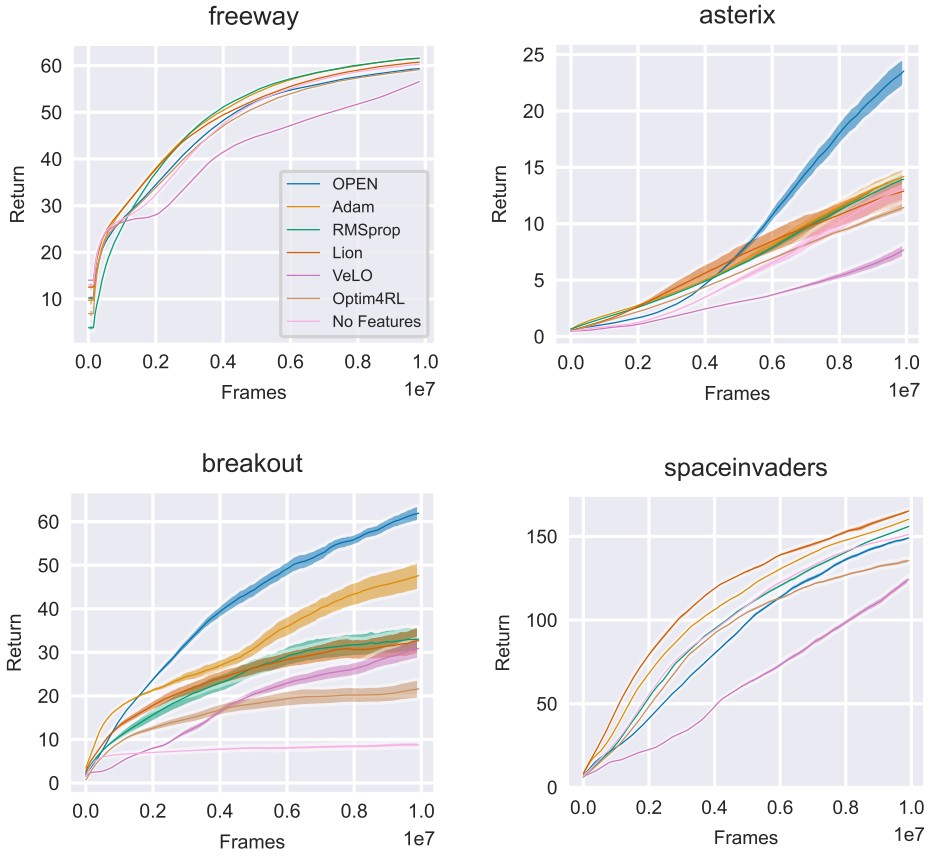

Figure 17: RL training curves for the multi-task setting. We show mean return over training with standard error, evaluated over 16 seeds.

# H   Additional Gridworld Experiments

Similar to Figure 4, which looked at how OPEN generalized to different agent sizes, we consider how OPEN adapts to different training lengths in Figure 18. In particular, we note that OPEN was only ever trained for gridworlds ('all', Appendix D) which ran for $3e7$ training steps, so all lengths outside of this are OOS tasks. We also consider that in E, the return of OPEN often started *lower* than other optimizers and increased only later in training. As such, the lifetime conditioning (i.e., training proportion) of OPEN needs to be leveraged to be able to perform well at *shorter* training lengths. If OPEN did not make use of this feature, it may end training without having converged to the nearby optima, instead focusing on exploration.

We find that OPEN is able to beat Adam in all of the grids considered at the *in-distribution* training length. For OOS training lengths, OPEN generalizes better than Adam, and only in a couple of environments (rand_dense, standard_maze) does it not outperform Adam in the shortest training length. For every other training length, OPEN performs better than Adam in *every* environment, further demonstrating its capability to generalize to OOS environments and training settings.

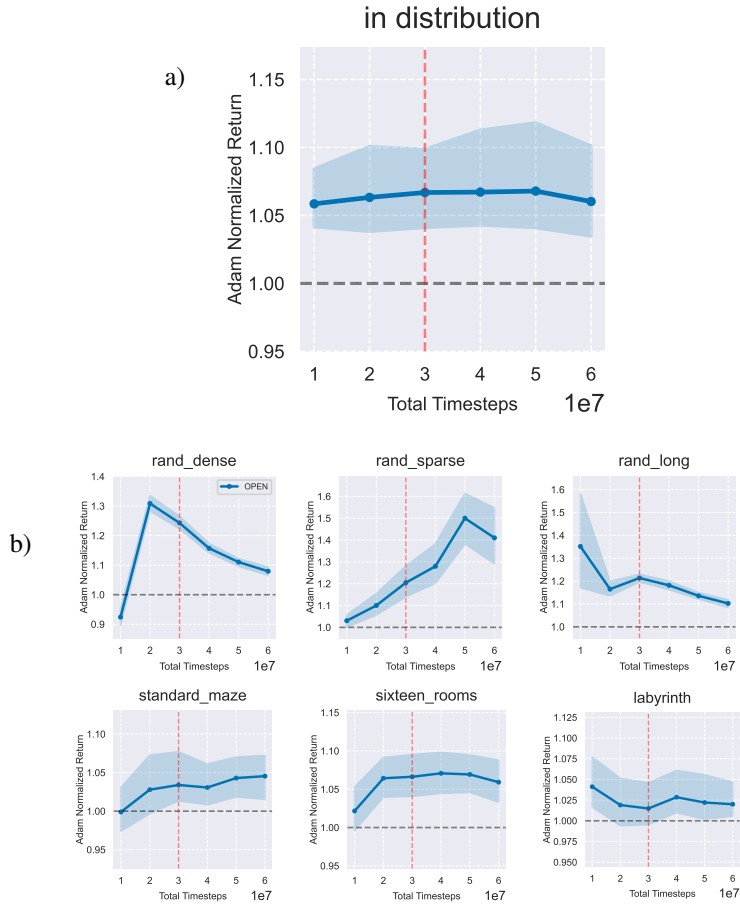

Figure 18: IQM of return achieved by OPEN, normalized by the return of Adam, against a range of different training lengths with 95% stratified bootstrap confidence intervals for 64 seeds. a) shows performance on the distribution which OPEN was trained on, and Adam was tuned in. b) shows performance on OOS gridworlds, with the top row coming from [13] and the bottom row inspired by mazes from [67], all implemented by [66]. We mark $3e7$ timesteps as the in-distribution training length for both OPEN and Adam.

# I Ablation Details

## I.1 Feature Ablation Setup

We train 17 optimizers for each feature ablation, and evaluate performance on 64 seeds per optimizer, using a shortened version of Breakout. We use the same PPO hyperparameters for Breakout as in Table 3 besides the total timesteps $T$, which we shorten to $2 \times 10^5$. Each optimizer takes $\sim 60$ minutes to train on one GPU, and we train a total of 119 optimizers.

We train each optimizer for 250 generations, keeping all other MinAtar ES hyperparameters from Table 9.

## I.2 Deep Sea Expanded Curve

We show the final return against size for the Deep Sea environment [71, 63] in Figure 16 for all optimizers; both shared and separated, with stochasticity and without. The one without can be thought of as a *deterministic optimizer*, and is denoted 'ablated' in the figure. For this experiment, we trained each optimizer for only 48 iterations on sizes in the range [4, 26]; any sizes outside of this range at test time are out of support of the training distribution.

Clearly the separated optimizer with learned stochasticity is the only one which is able to generalize across many different sizes of Deep Sea. This occurs marginally at the behest of optimization in smaller environments, where the stochastic optimizer still explores after obtaining reward and so does not consistently reach maximum performance. In Deep Sea, the final return scale is between $0.99$ and $-0.01$.

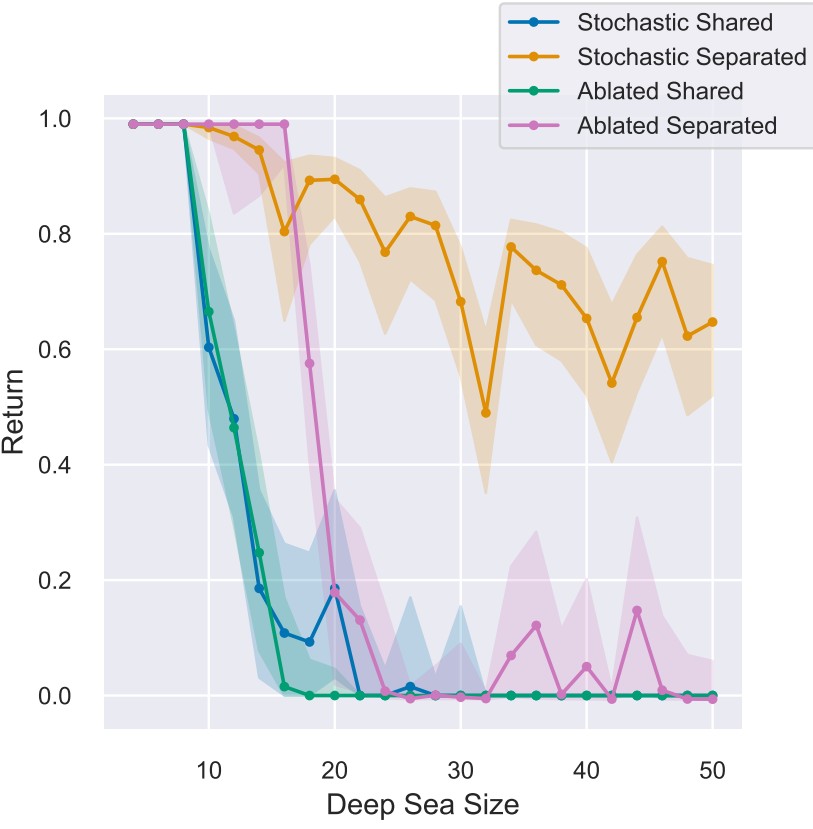

Figure 19: IQM of return against size of environment in Deep Sea, with stratified bootstrap 95% confidence intervals for 128 random seeds. The only optimizer which is able to generalize has separate parameters between the actor and critic and incorporates *learned stochasticity* into its update.

### I.3 PQN Return Curve

Figure 20 shows the performance of OPEN and Adam when training a PQN agent [72] in Asterix [62, 63]. While OPEN outperforms Adam in this setting, we note that it exhibits similar behavior to the other return curves (Appendix E, G) in having much lower return at the beginning of training before overtaking Adam at the end of training. As discussed in section 6.2, this suggests that OPEN sacrifices greedy optimization in favor of long-term performance.

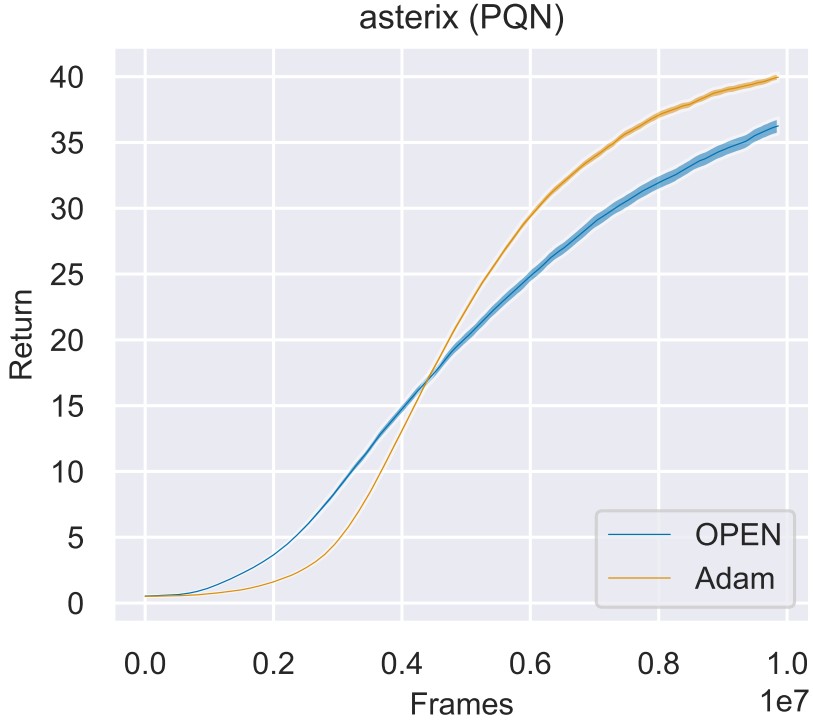

Figure 20: Training curves for PQN when optimised by OPEN and Adam in asterix. We show mean return with standard error, over 64 seeds.

## J Experimental Compute

### J.1 Runtimes

We include runtimes (inference) for our experiments with the different optimizers on $4 \times$ L40s GPUs. We note that, while OPEN takes longer to run compared to handcrafted optimizers, it offers significant speedup over VeLO and takes a similar runtime as Optim4RL and No Features in addition to significantly stronger performance. Importantly, this suggests the additional computation used in OPEN to calculate features, such as dormancy, does not translate to significant overhead compared to other learned optimizers.

Table 13: Runtime for each optimizer. We evaluate each over 16 seeds, on 4 L40s GPUs. These take advantage of Jax's ability to parallelize over multiple GPUs [33].

| Optimizer | RunTime (s) | | | | |
| --- | --- | --- | --- | --- | --- |
| | freeway | breakout | spaceinvaders | asterix | ant |
| OPEN *(single-task)* | 138 | 87 | 94 | 137 | 453 |
| OPEN *(multi-task)* | 154 | 98 | 109 | 148 | N/A |
| *Adam* | 91 | 53 | 58 | 105 | 333 |
| *RMSprop* | 84 | 38 | 52 | 97 | 322 |
| *Lion* | 83 | 44 | 50 | 96 | 303 |
| *VeLO* | 190 | 135 | 145 | 185 | 581 |
| *Optim4RL (single-task)* | 127 | 85 | 90 | 131 | 404 |
| *Optim4RL (multi-task)* | 168 | 99 | 116 | 157 | N/A |
| *No Features (single-task* | 123 | 79 | 88 | 132 | 416 |
| *No Features (multi-task)* | 154 | 93 | 106 | 147 | N/A |

### J.2 Training Compute

We used a range of hardware for training: Nvidia A40s, Nvidia L40ses, Nvidia GeForce GTX 1080Tis, Nvidia GeForce RTX 2080Tis and Nvidia GeForce RTX 3080s. These are all part of an internal cluster. Whilst this makes it difficult to directly compare meta-training times per-experiment, we find training took roughly between $\sim [16, 32]$ GPU days of compute per optimizer in the single-task setting, and $\sim 60$ GPU days of compute for multi-task training.

### J.3 Total Compute

For all of our hyperparameter tuning experiments, we used a total of $\sim 16$ GPU days of compute, using Nvidia A40 GPUs.

For each learned optimizer in our single-task experiments, we used approximately $\sim [16, 32]$ GPU days of compute. This equates to around 1 GPU year of compute for 3 learned optimizers on 5 environments (=15 optimizers total). This was run principally on Nvidia GeForce GTX 1080Tis, Nvidia GeForce RTX 2080Tis and Nvidia GeForce RTX 3080s.

For each multi-task optimizer, we used around 60 GPU days of compute, on Nvidia GeForce RTX 2080Tis. This totals 180 GPU days of compute. We also ran experiments for each optimizer with the smaller architecture, which took a similar length of time. These are not included in the paper; in total multi-task training used another 1 GPU year of compute, taking into account the omitted results.

To train on a distribution of gridworlds, we used 7 GPU days of compute on Nvidia GeForce RTX 2080 Tis.

Our ablation study used $\sim 8$ GPU days of compute, running on Nvidia A40 GPUs.

Each optimizer took 2 days to train in deep sea on an Nvidia A40 GPU. In total, this section of the ablation study took 8 GPU days.

As stated in Table 13, inference was a negligible cost (on the order of seconds) in this paper.

Totaling the above, all experiments in this paper used approximately **2**.**1 GPU years** to run, though it is difficult to aggregate results properly due to the variety of hardware used.

We ran a number of preliminary experiments to guide the design of OPEN, though these are not included in the final manuscript. We are unable to quantify how much compute was used to this end. Due to the excessive cost of learned optimization, we ran only one run for each setting and optimizer (besides the multi-task setting, where we tested both small and large architectures and found that the larger architectures performed best for every optimizer on average). After converging upon our method, and finishing the implementations of Optim4RL and *No Features*, we did not have any failed runs and so included all experiments in the paper.

# K   Code Repositories and Asset Licenses

Below we include a full list of assets used in this paper, in addition to the license under which it was made available.

Table 14: Main libraries used in this paper, links to the github repository at which they can be found and the license under which they are released.

| Name | Link | License |
|---|---|---|
| evosax [34] | evosax | Apache-2.0 |
| gymnax [63] | gymnax | Apache-2.0 |
| brax [64] | brax | Apache-2.0 |
| learned_optimization [45] | learned_optimization | Apache-2.0 |
| groove (gridworlds) [66] | groove | Apache-2.0 |
| purejaxrl [15] | purejaxrl | Apache-2.0 |
| optax [69] | optax | Apache-2.0 |
| rliable [70] | rliable | Apache-2.0 |
| craftax [21] | craftax | MIT |
| purejaxql (PQN) [72] | purejaxql | Apache-2.0 |

