# OpenReview forum: "Can Learned Optimization Make Reinforcement Learning Less Difficult?"
_NeurIPS.cc/2024/Conference — NeurIPS 2024 spotlight_

### Official Review · Reviewer_qjkS · 2024-07-06

**Soundness:** 3
**Presentation:** 3
**Contribution:** 3
**Rating:** 7
**Confidence:** 4

**Summary:**

This paper presents OPEN, a meta-learning approach with a learned optimiser. The approach is based on addressing three challenges for RL: 1) the non-stationary nature of input and output distributions; 2) the loss of plasticity; and 3) exploration.  The approach performs better than a traditional Adam optimiser and other learned optimisation techniques (Optim4RL, L2O, VeLO, and Lion), and works for both single- and multi-task environments.

**Strengths:**

The paper is clearly written and clearly highlights (and makes a good case for mitigating) general problems inherent in the RL framework, namely non-stationarity, plasticity loss, and exploration. The background and definitions are clear.

The technique uses per-parameter learned parameter noise to both increase exploration and reduce loss of plasticity. The optimiser update rules are learned based on evolutionary strategies taking dormancy, layer proportion, training proportion, and batch proportion as inputs, which is a novel approach as far as I am aware.

The approach works well compared to state-of-the art and provides a scalable approach.

Using the approach, the exploration parameter is increased automatically for large environments, which is intuitively what you would expect.

The study takes care of ablations to verify the impact of each of the parameters in the meta-optimisation.

**Weaknesses:**

The techniques used for the empirical comparisons are not described in enough detail.

The review of directly related techniques is limited, even though I am aware that this is a niche domain.

One of the key components is parameter noise. It is not clear how the paper compares to approaches for parameter noise, e.g. citation [11] in your paper. It may for instance be possible that parameter noise alone is beneficial. This is important to verify but I currently do not see any such analysis in the paper.

The paper tries to address different points at the same time and there are many components. In a sense, the contribution of each individual part is limited and the technique is a "combination" of techniques rather than a new technique altogether.

**Questions:**

l.75 is use --> is used

The automated exploration rate tuning based on the environment and meta-learning of algorithmic steps is similar to Active Adaptive Perception [1]. It is relevant to review it, e.g. in the “Discovering RL algorithms” subsection.

Appendix E. Why is Ant being evaluated for more time steps than the other conditions?

It would be good to see an ablation comparing your system with citation [11] or other techniques for parameter exploration. It is not clear whether there is an algorithmic improvement in terms of the parameter exploration rule itself.

------

[1] D.M. Bossens, N.C. Townsend, A.J. Sobey (2019). Learning to learn with active adaptive perception, Neural Networks,
115, 2019, Pages 30-49.

**Limitations:**

The paper describes limitations in terms of bias towards environments where Adam underperforms, the limits of the scope of the study in terms of meta-optimisation (although I think the scope of meta-optimisation is already large), and the base-learner used.

In addition, I think there is an additional limitation that should be mentioned. The study focused on gridworlds. It would be interesting to see how the technique scales to high-dimensional problems.

---

> ### Author Rebuttal · Authors · 2024-08-05
>
> Dear reviewer qjkS,
>
> Thank you for your in-depth review and recommendation of acceptance. We greatly appreciate that **you find our approach performant, scalable and novel**. We address your concerns below.
>
> # Lack of Empirical Detail
>
> >The techniques used for the empirical comparisons are not described in enough detail.
>
> We appreciate your feedback, but wondered if you could please clarify which details in particular are missing from our experimental desciptions? In our work, we:
> - Explain each training environment and task setting thoroughly.
> - Discuss a range of baselines, with hyperparameters and tuning details (Appendix C).
> - Present results in a standardized framework (rliable), reporting 95% stratified bootstrap CIs over many seeds.
> - Provide conclusions in text which can be drawn directly from figures, without extraneous claims.
> - Include extra details in the appendix, including return curves (Appendix E, G), analysis (Appendix F) and more results (Appendix H).
> - Ablate each component of OPEN, considering the impact on final return and the agents' dormancy, with more details in Appendix I.
> - Ensure all sections are properly cross-referenced and adequately cited.
>
> # Review of Directly Related Techniques
>
> > The review of directly related techniques is limited, even though [...] this is a niche domain.
>
> Please could you highlight what specifically is missing from our review of related techniques? We thank you for, and have included into our related work, your suggested reference on "Active Adaptive Perception". However, regarding the overarching point, we:
> - Include extensive background, covering evolution and optimization from both a handcrafted and learned perspective.
> - Have a detailed related work section introducing meta-learned algorithms in RL, such as LPO, LPG and MetaGenRL, and learned optimization in non-RL (e.g. VeLO, Lion). We also describe Optim4RL.
> - Dedicate a section to three difficulties of RL, with ample links to literature. This is expanded in Appendix A.
> - Motivate many design decisions in our optimizer from prior literature.
> - Compare OPEN against Optim4RL, VeLO and Lion, which are all "directly related techniques" involving learned optimization. We also compare against Adam and RMSProp, which are commonly used in RL.
>
> # Comparison to Parameter Space Noise
>
> > It is not clear how the paper compares to approaches for parameter noise
>
> Directly swapping out learnable stochasticity for parameter space noise is, unfortunately, infeasible. Parameter space noise for exploration introduces *two new hyperparameters*: how much noise to add; and the threshold when noise scales. To tune these while learning to optimize would be excessively costly, as each hyperparameter run would take on the order of days.
>
> However, our analysis (Appendix F.4, F.5) shows that OPEN learns stochastic behaviour that **could not be represented with vanilla parameter space noise**, like producing higher levels of stochasticity in larger Deep Sea environments *without awareness of the environment size*. It is important to note that our learnable stochasticity conditions on *all of OPEN's inputs*, meaning there are complex interactions between all of our features and the stochasticity. Therefore, it is unlikely that vanilla parameter space noise could replicate our performance.
>
> Interestingly, Bigger, Better, Faster [1] actually removes noisy nets, which are similar to learnable stochasticity in OPEN, due to over-exploration. Our approach, which meta-learns how much noise to apply training, improves performance, **suggesting OPEN resolves this issue**.
>
> We have now made clear in the paper the advantages of OPEN over parameter space noise.
>
> # Minor Contribution From Individual Elements
>
> >the contribution of each individual part is limited and the technique is a "combination" of techniques rather than a new technique altogether.
>
> We believe our work offers significant contribution as the **first to show learned optimization consistently outperforming baselines in RL**. While we refer you to the expanded list in our global response, our key contributions include:
>
> - Learning to optimize in RL *using evolution strategies*, which has only recently made feasible thanks to JAX acceleration.
> - Producing a small number of input features, verified via ablation, that improve the performance of the learned optimizer without requiring excessive hand-designed structure. These features are *grounded* in theory, but are derived novelly.
> - Introducing *meta-learned exploration* directly in the optimizer, using learnable stochasticity.
> - Stabilizing training with zero-meaning. This does not excessively limit the expressiveness of the optimizer but enables meta-training in environments with continuous actions.
>
> While OPEN has several components, it is a mischaracterization to reduce it to only a 'combination of techniques'. Our method **significantly advances the SOTA in learned optimization for RL**. We also note that many successful papers integrate multiple findings into a single method, such as [1].
>
> # Limiting to Gridworlds
>
> >It would be interesting to see how the technique scales to high-dimensional problems
>
> In response to your concerns regarding the gridworld experiments, we have added new results in a much harder domain: Craftax-Classic, a JAX reimplementation of Crafter. We provide results and extra detail in the general response.
>
> In short, we demonstrate **better 0-shot generalization** than Adam from our distribution of gridworlds to Craftax, and **perform only slightly worse, 0-shot, than Adam when it is finetuned in domain**.
>
> # Specific Questions
>
> Thank you for spotting a typo, we have updated the paper. For Ant, our hyperparameters based on prior work which runs in Brax ($5\times10^7$) for longer than MinAtar ($1\times10^7$). Training for more timesteps on Brax than MinAtar is common.
>
> ---
>
> [1] Max Schwarzer et al., Bigger, Better, Faster: Human-level Atari with human-level efficiency. 2023

---

> ### Comment · Reviewer_qjkS · 2024-08-09
>
> Thanks for the response, I still have a few comments.
>
> The new results look promising at first glance but we would need to see the spread across runs to indicate the confidence in the result. The authors mention the standard error but either there is a mistake or it is too small. Perhaps the standard deviation makes more sense if it is too small to see.
>
> > Directly swapping out learnable stochasticity for parameter space noise is, unfortunately, infeasible. Parameter space noise for exploration introduces two new hyperparameters: how much noise to add; and the threshold when noise scales. To tune these while learning to optimize would be excessively costly, as each hyperparameter run would take on the order of days.
>
> This was not the question. The question was how your approach compares to RL with parameter noise, not meta-RL with parameter noise.
>
> > However, our analysis (Appendix F.4, F.5) shows that OPEN learns stochastic behaviour that could not be represented with vanilla parameter space noise, like producing higher levels of stochasticity in larger Deep Sea environments without awareness of the environment size. It is important to note that our learnable stochasticity conditions on all of OPEN's inputs, meaning there are complex interactions between all of our features and the stochasticity. Therefore, it is unlikely that vanilla parameter space noise could replicate our performance.
>
> Actually, the claim does not necessarily follow from these plots. It could simply be the case that (due to dividing by the parameter value) the noise remains constant but the parameter value is changing all the time. The larger environment size might be correlated with smaller parameter values.
> There is also no comparable plot for the vanilla parameter space noise case.
>
> > We believe our work offers significant contribution as the first to show learned optimization consistently outperforming baselines in RL. While we refer you to the expanded list in our global response, our key contributions include:
>     • Learning to optimize in RL using evolution strategies, which has only recently made feasible thanks to JAX acceleration.
>     • Producing a small number of input features, verified via ablation, that improve the performance of the learned optimizer without requiring excessive hand-designed structure. These features are grounded in theory, but are derived novelly.
>     • Introducing meta-learned exploration directly in the optimizer, using learnable stochasticity.
>     • Stabilizing training with zero-meaning. This does not excessively limit the expressiveness of the optimizer but enables meta-training in environments with continuous actions.
> While OPEN has several components, it is a mischaracterization to reduce it to only a 'combination of techniques'. Our method significantly advances the SOTA in learned optimization for RL. We also note that many successful papers integrate multiple findings into a single method, such as [1].
>
> I think that these points of novelty are not clear from the text, for instance, these points are not highlighted in the intro.  Also the novelty and mechanism of each of these points is difficult to expand upon (and therefore evaluate) seriously due to having so many components. Whether or not it advances SOTA performance is not directly related to whether it is a combination of techniques or not. That said, this is not a point of rejection for me and I believe that many methods can be characterised like this – some more easily than others.

---

> ### Author Response · Authors · 2024-08-12
>
> Dear qjkS,
>
> Thank you for your response and continued engagement in the rebuttal process. We apologize for the delay in our response; we have spent some time generating additional results.
>
> # Craftax Results
>
> If you zoom in to the plot, there are standard error intervals, but as in the original Craftax paper these are very small. For your benefit, we provide the specific values for final return below:
>
> | Optimizer          | Mean ± Std Err       |
> |---------------------|-------------------|
> | Adam (Finetuned)   | 13.842 ± 0.037  |
> | OPEN (0-shot)      | 12.066 ± 0.036  |
> | Adam (0-shot)             |  7.490 ± 0.053  |
>
> # Additional Baselines
>
> We apologize for misunderstanding your original review point. We have now implemented Noisy Networks for Exploration [1] with PPO on the MinAtar environments. This allows for a *learned per-parameter variance* which is comparable to the noise applied by OPEN, rather than parameter space noise (where noise variance is shared across the whole agent). We try to follow the implementation of noisy networks for A3C, though highlight a few important implementation details and differences below:
> - We maintain a constant noise *between updates* rather than over episodes, as would be standard; in JAX, with vectorized environment rollouts, it is far from trivial to have constant episodic noise unless episodes are of a constant length (which is not the case in MinAtar). OPEN also receives new noise every update rather than episodically.
> - We tune Adam (*LR*, $\beta_1$, $\beta_2$) for each MinAtar environment.
> - Unlike OPEN, where noise is *sticky* (i.e., maintained across updates), in noisy networks the noise gets resampled regularly.
> - In OPEN, the learnable stochasticity is applied *in addition* to the entropy bonus, whereas in noisy networks the noise is applied *instead of* entropy. In our noisy networks implementation, we remove the entropy bonus from PPO to align to literature.
> - We use the standard initialization of variance in noisy networks (0.017).
>
> Below, we provide results for PPO + noisy networks alongside the scores for Adam and OPEN.
> We report the IQM and 95% stratified bootstrap confidence intervals (min, max) for 16 seeds, following the reporting procedure from the paper.
>
> | Optimizer                  | Freeway                  | Breakout                 | Asterix                  | Space Invaders           |
> |-------------|--------------------------|--------------------------|--------------------------|--------------------------|
> | OPEN        | **64.42 (64.06, 64.74)**     | **66.94 (60.60, 72.02)**     | **36.84 (32.11, 39.96)**     | **167.22 (157.47, 175.76)**  |
> | Adam        | 62.26 (61.93, 62.58)     | 47.82 (37.44, 55.81)     | 14.97 (13.30, 16.42)     | **165.42 (162.71, 168.52)**  |
> | Noisy Networks (+Adam)  | 54.02 (53.47, 54.59)     | 38.09 (30.36, 48.47)     |  9.75 ( 5.46, 14.38)     | 124.86 (121.58, 129.58)  |
>
>
> We note that applying noisy networks, or any form of parameter noise, is uncommon with PPO; as a result, the fact that the more canonical implementation of PPO outperforms the noisy networks should not prove surprising. It is, however, interesting that learnable stochasticity provides such a benefit to OPEN; as we highlight in our paper, it may be that improvements arise from **both** improved exploration *and* reduced dormancy. It is also possible that it is significantly easier to *meta-learn* a noise profile than learn how much noise to apply *online* (as in noisy networks, where variance is an optimization parameter learned by RL).
>
> # Novelty
>
> We appreciate your comments around this, and have included some sentences describing the contributions of our method into the introduction.
>
> To evaluate each contribution, we provide thorough ablation of each of these components and find that each contributes to the overall performance of OPEN. The novelty of these contributions should not be difficult to interpret; no one has trained optimizers in RL with ES, our stabilization method has not been applied before, and while our inspiration for the various features and randomness is included in the paper, their incorporation into a meta-learned system is novel besides temporal conditioning, which was inspired by [2] (though we implement it slightly differently). In our original paper, we attempted to make this clear, but our new sentences in the introduction make this more concrete.
>
> We hope we have answered your concerns, but please do send a comment if you have additional questions that we can answer before the deadline.
>
> ---
>
> [1] Meire Fortunato et al., Noisy Networks for Exploration, 2017
>
> [2] Matthew Thomas Jackson et al., Discovering Temporally-Aware Reinforcement Learning Algorithms, 2024

---

### Official Review · Reviewer_KVk8 · 2024-07-12

**Soundness:** 4
**Presentation:** 4
**Contribution:** 4
**Rating:** 8
**Confidence:** 3

**Summary:**

This paper presents OPEN, a meta-learned optimizer for reinforcement learning gradient updates. The update rule for OPEN is split into 3 stages, one for each difficulty in RL they define: non-statiarity, plasticity loss and exploration. They train their optimizer with an evolutionary strategy. To analyze their results they run experiments on different setups with varying training and deployment domains. They also perform an ablation study to verify their choises. They find that their method significantly outperforms their baselines (mainly Adam), and is able to generalize well to different environments.

**Strengths:**

Quality: Major. The paper is sound tehcnically and their ablation study/experimental results are very well done. They are very thorough in their testing.

Clarity: Major. The paper usees bolding and italics exellently to provide roadmaps and make it easy to read. The ablations study plots are extremely easy to interpret. Information from the text and appendix should be enough to reproduce all results.

Significance: Above average. The results seem to be promising to improving RL algorithm performance only by improving the optimizer. Other than that, on the tasks they do experiment with their results show a large improvement over all baselines and seem to provide a step toward better RL performance.

Originality: Slightly below average. The high level idea here (learning an optimizer for RL) is not a novel idea, being done in the Optim4RL work. I don't think that this high level novelty is the main contribution of the paper and they don't claim it to be. They do take a different approach to learning the optimizer which is their novelty. Works are accurately cited.

**Weaknesses:**

I wish it was a little more clear how this work is novel (from the authors perspective). From my reading I gathered that it is the use of ES and specific identification/use of RL-specific challenges but a sentence or two in the intro would be fantastic.

They only apply their method to gridworld tasks, while this is completely reasonable due to possibly limited compute I would like to know if this generalizes to more complex tasks, or at least if the authors believe it will.

**Questions:**

If you feel your work is more novel than I am giving credit for this is something I am very willing to change my mind on if I'm wrong.

Your optimizer seems to generalize well, is it possible to train it once on a wide variety of environments and then freeze it and use it for basically any environment?

You address this in the appendix, but it would be nice to put in the main text a small sentence or two about how your method isn't much slower than Adam. That was a concern of mine as far as applying this if it is magnitudes slower than Adam that cancels out some benefit but it doesn't seem like it is.

**Limitations:**

Limitations: Maybe this is implied somehwere or I missed it, but applying this to more complex domains would be an interesting future work in my opinion. If you feel that your environments generalize to complex domains then this is not a limitation. None other than that.

---

> ### Author Rebuttal · Authors · 2024-08-05
>
> Dear reviewer KVk8,
>
> Thank you so much for your very positive review and recommendation of strong acceptance to the conference. While you have **highlighted many positives of our work, such as the quality of our writing, experiments and ablations**, you do still discuss some weaknesses of our paper. We address these below.
>
> # Novelty of the work
>
> > I wish it was a little more clear how this work is novel (from the authors perspective)
>
> We list below some key novel concepts we introduce in our paper, and refer to the general response for a more full list. We have now included a couple of sentences into the introduction clearly detailing them:
>
> - We use ES to learn to optimize *for RL*, which has only recently been made possible thanks to JAX acceleration.
> - We propose a way to design a black-box meta-learner in a *very unconstrained way* using input features. This lets the meta-learner *learn what's important* for the goal we care about, such as final return, rather than having regularization penalties for things such weight magnitude. Our method ensures we maximize the true objective, rather than an augmented one, while still accounting for other problems (like plasticity loss) as a byproduct.
> - We highlight a set of *theoretically grounded* features which can be combined to reduce the effect of some specific difficulties in RL. In addition to showing the importance of these features, via ablation, in learned optimization for RL, we believe that similar approaches can and should be applied to a plethora of learned algorithms for RL, and that our features should be generally universal.
> - We introduce a new meta-learned exploration technique which boosts exploration *directly through the learned optimizer*. We demonstrate that this is useful in a hard-exploration environment, DeepSea.
> - We introduce a way to stabilize learned updates in domains with continuous action spaces by zero-meaning the updates, which prevents overflow issues from exponentiation. This does not hurt empirical performance in already stable environments, and makes training possible in environments with continuous action spaces. According to the Optim4RL ICLR24 rebuttals (which can be found online), this was also an issue in a naive baseline they used and was one of the reasons they heavily constrained their update expression, limits its expressibility.
> - We are the first example of learned optimization in RL **which significantly outperforms all baselines in a range of domains**, including beating Adam consistently in generalization to different environments and to different architectures. In our general response and below, we provide even stronger results from our rebuttal experiments, which demonstrate very strong generalization from gridworlds to a harder environment (Craftax).
>
> # Gridworlds and Generalization
>
>
> > I would like to know if this generalizes to more complex tasks
>
> In our paper, we do outline our hopes of more complex generalization on line 285. It is our belief that the combination of our multi-task experiments, which show that OPEN can fit to multiple harder and diverse environments, and the generalization experiments, which look at generalizing from easy gridworlds to significantly harder gridworlds, demonstrate that more complex generalization results will arise from larger scale training.
>
> As part of our rebuttals we have considered transfer to *Craftax Classic* [1], a recent JAX reimplementation of Crafter. This is a **significantly harder environment than those tested in our paper**. Notably, since we are considering generalization rather than training directly in craftax, **we do not learn any new optimizers** for this experiment; instead, we evaluate the optimizer trained in section 6.4 of the paper on Craftax, 0-shot. We refer to the general response for full experimental details and results.
>
> In short, we find that OPEN **generalizes significantly better than Adam** from the gridworlds (i.e., if we also transfer the Adam hyperparameters tuned in the gridworld). Additionally, the 0-shot generalization performance of OPEN is *only slightly worse than an in-domain, finedtuned implementation of Adam*.
>
> This experiment suggests that OPEN is able to generalize very far out of distribution, and **offers the potential beginnings of truly automated RL** - in particular, meta-training can be done on simpler and quicker environments without the need for any finetuning on the hard/real-world environment!
>
> > is it possible to train it once on a wide variety of environments and then freeze it and use it for basically any environment
>
> As an academic lab, it is beyond our capabilities to thoroughly explore the question of large-scale training. However, given the sheer levels of generalization we have seen from just our small-scale experiments, and the newfound capability to generalize well to Craftax-Classic, **it is our belief that OPEN could be trained once on a variety of environments and then be applied near-universally in RL**. We think that this would be a natural and highly impactful next step to OPEN and would highly recommend others with greater compute capabilities to take this forward, making use of our open source codebase. We have included a statement to this effect in the paper.
>
> # Training Speed
>
> > it would be nice to put in the main text a small sentence or two about how your method isn't much slower than Adam
>
> Thank you for your suggestion about better communicating the speed of training with OPEN. We have included a sentence highlighting that OPEN does not drastically slow down training; in fact, since the main bottleneck in training RL is generally the environment interaction, it is likely that the slowdown from OPEN will decrease as it scales to more complex settings.
>
> ---
>
> [1] Michael Matthews et al., Craftax: A Lightning-Fast Benchmark for Open-Ended Reinforcement Learning. 2024.

---

> > ### Comment · Reviewer_KVk8 · 2024-08-08
> >
> > Thank you for a response to my questions. The idea to possibly train on a variety of environments and universally apply the optimizer is one that I believe has far reaching implications. Great job.

---

> > > ### Author Response · Authors · 2024-08-12
> > >
> > > Dear KVk8,
> > >
> > > Thank you for your additional response, and we are very grateful that you appreciate our work! Please do let us know if you have any other queries about our work that we might be able to answer.

---

### Official Review · Reviewer_Fr37 · 2024-07-18

**Soundness:** 3
**Presentation:** 4
**Contribution:** 3
**Rating:** 7
**Confidence:** 4

**Summary:**

This paper explores whether learned optimization can address specific challenges in reinforcement learning (RL), such as non-stationarity, plasticity loss, and the need for exploration. The authors propose an optimizer, OPEN, which meta-learns update rules related to these issues. OPEN demonstrates strong performance and generalization across various environments compared to traditional and other learned optimizers. The study confirms that learned optimizers, when well-parameterized and trained, can improve the PPO algorithms' effectiveness.

**Strengths:**

- Very clear and well-written paper.
- Comprehensive Introduction and Related Work.
- Strong evaluation using Adam-Normalized scores.
- Extensive ablation studies.

**Weaknesses:**

- Line 135: Plasticity loss. The use of ReLU dormancy is, in my opinion, not a viable metric for plasticity in RL. Although interesting work, [1] has shown that combatting ReLU dormancy does not improve 'vanilla' nature DQN, unless you increase the replay ratio. Further work [2] has also shown that combatting dormant neurons as in [1] significantly reduces Impala DQN performance, while extensive network pruning increases performance. This raises the question of whether neuron dormancy is actually related to plasticity in RL.

- An evaluation on another RL algorithm or a more complex environment (Atari?) would strengthen the results. (The PureJaxRL Github Repo [3] also has DQN)

Small remarks ##:

- Line 73: "Evolution algorithms" -> 'Evolutionary algorithms'
- Line 75: "This population is use" -> 'This population is used'
- Line 250: "significantly outperform" -> 'significantly outperforms'



[1] Sokar et al: The Dormant Neuron Phenomenon in Deep Reinforcement Learning

[2] Obando-Ceron et al: In value-based deep reinforcement learning, a pruned network is a good network

[3] Lu et al: Discovered policy optimisation

**Questions:**

Can the authors say something about the computational cost of their method versus the other methods used in their evaluation?

**Limitations:**

Related to my question, maybe a small section about computational differences should be added to the main text.

---

> ### Author Rebuttal · Authors · 2024-08-05
>
> Dear Fr37,
>
> Thank you very much for your review. We are glad that you appreciate our **clear writing and strong evaluation** throughout the paper, and were happy to see this reflected in your recommendation of acceptance. Below, we detail answers to your queries.
>
> # Insufficiency of Dormancy
>
> > The use of ReLU dormancy is, in my opinion, not a viable metric for plasticity in RL.
>
> As we cite in our paper, there exists a long list of work which use dormancy as a proxy measure for plasticity loss; e.g., [1,2,3,4], to name a few. This would suggest that, though dormancy and plasticity are not *directly* equivalent, dormancy is a useful and easily computed metric emblematic of plasticity. While the works you cite are certainly interesting, it is important to recognise that some of these findings (about e.g., the replay ratio) **do not apply in our setting**, where we have thus far focused on PPO.
>
> More importantly, however, is the fact that we run our *own experiments* in the paper which demonstrate the **significant benefit of providing dormancy directly to the optimizer**; in our ablation (Figure 5), which is evaluated over a large number of optimizers, we find that dormancy was the **most significant individual feature** input to OPEN. In addition to causing a ~20% drop in final return, the percentage of dormant neurons tripled when dormancy was not included at input. Considering the full-scale experiments, OPEN also generally produced fewer dormant neurons despite optimizing *only for final return* (Appendix F1).
>
> These results directly lead to 2 conclusions:
> - Including dormancy at input to the optimizer significantly improves final return.
> - Since OPEN (with awareness of dormancy) produced significantly less dormant neurons when optimizing for final return, dormancy must have a direct link to performance in this setting.
>
> While we believe, as grounded in literature, that the primary reason for these two effects comes down to the effect of plasticity, it may be that there are additional effects beyond our understanding that causes improved performance by reducing dormancy. However, our experiments make clear that **dormancy is an important input feature to OPEN which provides significant benefit**.
>
> Based on the above, we hope that you agree that there is a close link between dormancy and performance in our setting, and that there is existing literature beyond our work by which we can support the claims made in our paper.
>
> # Considering Harder Environments and Other Algorithms
>
> > evaluation on another RL algorithm or a more complex environment (Atari?) would strengthen the results
>
> In response to your question, we have run experiments focusing on generalization from gridworlds to a **much more complex environment**. In our global response, we introduce an additional generalization experiment on Craftax-Classic [5], a recent reimplementation of Crafter in JAX. In this setting, we find that OPEN is able to **generalize significantly better than Adam** from the gridworlds (i.e., if we also transfer the Adam hyperparameters tuned in the gridworld). Also, the 0-shot generalization performance of OPEN is *only slightly worse* than an in-domain, finedtuned implementation of Adam.
>
> We refer the reviewer to the global response and PDF for more details.
>
> Unfortunately, due to how long it takes to train a learned optimizer, as well as server issues, we have been unable to finish training OPEN with another RL algorithm in the allotted rebuttal time; **this experiment is therefore still ongoing**.
>
> We are currently training OPEN with PQN, a vectorized version of DQN. After extensive finetuning, Adam + PQN achieves ~36.5 return on Asterix against ~34.9 return for OPEN (16 seeds). However, **we anticipate OPEN will eventually overtake Adam since the meta-training performance is still consistently increasing**. We will update you with a comment later in the week, pending the experiment's conclusion.
>
> # Computational Cost
>
> > Can the authors say something about the computational cost of their method versus the other methods used in their evaluation?
>
> We include a very thorough breakdown of evaluation and training cost in appendix J. They key takeaway is that OPEN is significantly quicker than VeLO and not that much slower than Adam in test-time training, with the discrepancy shrinking in more expensive environments where the simulation becomes the bottleneck. In meta-training, OPEN generally converges faster than *No Features* or Optim4RL; we include details of how many training generations each one used in Appendix C.3.
>
> We apologize for not having included sufficient reference to this in the main body of the text. We have added additional references in an updated manuscript, alongside a sentence explaining the computational differences between OPEN and other optimizers.
>
> ---
>
> [1] Clare Lyle et al., Disentangling the causes of plasticity loss in neural networks, 2024.
>
> [2] Hojoon Lee et al., PLASTIC: Improving Input and Label Plasticity for Sample Efficient Reinforcement Learning. 2023.
>
> [3] Guozheng Ma et al., Revisiting Plasticity in Visual Reinforcement Learning: Data, Modules and Training Stages. 2023.
>
> [4] Zaheer Abbas et al., Loss of Plasticity in Continual Deep Reinforcement Learning. 2023.
>
> [5] Michael Matthews et al., Craftax: A Lightning-Fast Benchmark for Open-Ended Reinforcement Learning. 2024.
>
> [6] Matteo Gallici et al., Simplifying Deep Temporal Difference Learning. 2024.

---

> ### Comment · Reviewer_Fr37 · 2024-08-08
> **Response to Rebuttal**
>
> Dear authors,
>
> Thank you for your extensive rebuttal.
>
> # Dormancy
> I agree with your conclusion that dormancy is very likely related to performance. However, I tend to see dead neurons more as a symptom of poor optimization, rather than having a causal relationship to plasticity. I think your use of dormancy as an OPEN feature allows it to treat the disease rather than treat the symptoms. But I might be wrong :).
>
> # Harder Environments
> Thanks for adding the Craftax environment. Even though the consensus already seems to lead to acceptance, this will certainly strengthen the paper.
>
> # Computational cost
> A more visible reference to the computational cost appendix will be useful, thanks.
>
>
> Given that the authors have done an extensive rebuttal, and are committed to adding an additional environment in their evaluation, I have increased the score to a 7.

---

> ### Author Response · Authors · 2024-08-12
>
> Dear Fr37,
>
> Thank you for your detailed review and engagement in our rebuttal; we are very grateful that you have increased your score.
>
> Since our experiment has concluded, we now share our final results comparing OPEN against Adam for PQN [1], a parallelised and simplified implementation of DQN designed for JAX. Here, we slotted OPEN *directly* into the PQN code available on github, with no algorithmic changes made to OPEN.
>
> Due to compute limitations, we were only able to train OPEN from a single seed in one environment, Asterix, though we see no reason that these results would not transfer to other environments. As was standard in our work, we used many hyperparameters directly from the PQN paper but extensively tuned Adam (*LR*, $\beta_1$, $\beta_2$, *Anneal_LR*) with 8 seeds per hyperparameter. We are happy to give full hyperparameters if desired.
>
> Our results are as follows, assessed over 64 seeds with standard error. OPEN significantly outperforms Adam.
>
> | Optimizer    | Final Return ± Std Err        |
> |---------------|-------------------|
> | Adam          |   36.86 ± 0.68  |
> | OPEN    |   **40.06 ± 0.42**  |
>
> While I am unable to show you our return curve due to the limit of a 1 page PDF submitted during the rebuttal period only, we also observe some interesting behaviour similar to that of OPEN with PPO; OPEN only overtakes Adam later in training. This suggests that many of our findings about the behaviour of OPEN with PPO also hold with other algorithms. However, further analysis would be required to verify this, which we have not had the time to carry out during the rebuttal period.
>
> We hope that this satisfies your concerns regarding other algorithms. We will ensure to update our paper with these additional results.
>
> [1] Matteo Gallici et al., Simplifying Deep Temporal Difference Learning. 2024.

---

### Official Review · Reviewer_sKXo · 2024-07-25

**Soundness:** 3
**Presentation:** 3
**Contribution:** 3
**Rating:** 7
**Confidence:** 4

**Summary:**

This paper presents OPEN, which meta-learns an update rule (a learned optimizer) for reinforcement learning. The authors test in single-task and multi-task settings, and show good generalization to tasks outside of the training set.

**Strengths:**

This is a good paper on the application of learned optimization to RL. Of particular note:
- I like the focus on learning capacity via single-task learning. This is a good sanity check for learned optimizers.
- The out-of-distribution experiments are valuable.
- Discussion of acceleration via parallelization in JAX is an extremely important engineering detail.
- The explicit conditioning on dormancy is an interesting contribution. There is interesting potential future work within the learned optimization community on useful theoretically-driven inputs to learned optimizers.

**Weaknesses:**

- The zero mean last step (eq (9)) seems surprising for an update. Indeed, it seems quite different compared to eg momentum or other normalization strategies (such as adam-style normalization). Better ablations on this term would be helpful. It makes sense that adding random noise at each timestep could destabilize the model (the variance as t -> \infty will grow without limit), but could you instead use an additive noise process with a contraction term that has bounded or shrinking variance instead?
- The notation in eq (8) is a bit hard to parse---it reads more like an assignment in CS than an equality. Even though a def eq is used here, I think the notation could be improved for clarity.

**Questions:**

- How does the OpenAI ES strategy compare to the ES strategies used in the previous work on learned optimization, like standard ES, PES, NRES, etc?
- Are there alternate methods for preventing parameter growth from the exploration term?

---

> ### Author Rebuttal · Authors · 2024-08-05
>
> Dear sKXo,
>
> We would like to thank you for your positive review of our paper. In particular, we are grateful that you appreciate our **contribution based on conditioning directly on dormancy and other theoretically-grounded inputs** and recognize that this opens up huge potential opportunities in learned optimization. Below, we detail specific answers to the queries raised in your review.
>
>
> # Zero-Mean Updates Seems Counterintuitive
>
> > it seems quite different compared to eg momentum or other normalization strategies (such as adam-style normalization)
>
> It is worth noting that OPEN **does use momentum** (as detailed in Appendix B.1, which discuss that different timesteps of momentum are included at input to OPEN).
>
> > It makes sense that adding random noise at each timestep could destabilize the model
>
>
> It is important to highlight that **the instability is not caused by the inclusion of randomness**. We previously ran experiments with and without randomness, and found that the instability persisted until we incorporated the zero-meaning component. In fact, the ICLR24 rebuttals for Optim4RL are openly available online (I am not permitted to post a link per NeurIPS rebuttal guidelines) and **discuss a similar effect with their "linear optim" baseline, where they had NaN issues when learning in Brax** (quote: "Note that LinearOptim in Ant fails due to NaN errors"). They have little discussion of why this occurs, but we believe their issues arise due to the same problem as we discuss.
>
> Based on our analysis, we discuss the root cause of this issue in our paper (line 173); learned optimizers generally seem to produce large parameters. In domains with continuous action spaces, like Ant, the actor has an output parameter corresponding to the $\log(\sigma)$ of the actions (which gets exponentiated). Given that Brax also trains on long horizons, there is ample opportunity for parameters to grow and cause overflow errors.
>
> >  Better ablations on this term would be helpful
>
> Rather than the interesting examples of normalization you propose, such as Adam-style (similar to Optim4RL and heavily constrains the model expressiveness) or contractive terms in the noise (which is insufficient, as the noise does not cause the instability), we have previously tested a regularization penalty in the fitness (i.e., $\text{Fitness} = \text{Final Return} - \lambda||\theta||_2$, where $\theta$ is the agent's parameters). However, this introduces an additional, very expensive to tune hyperparameter ($\lambda$) and frequently failed to escape the local minimum at $\theta\approx0$. Since these results are trivial, and tuning $\lambda$ for a learned optimizer is beyond our capabilities, we **do not believe these results need to be included in the paper**. Instead, we have added mention of them in the main text to aid others and prevent dead-ends in their future research.
>
> We found that zero-meaning did not significantly limit the flexibility of the update rule, does not require careful hand-design, and empirically had no impact in environments where exponentiation does not cause issues while making training possible in those it does. As such, we hope that you agree that **no additional ablations are needed on this term**, though we have now mentioned in the paper that this instability occurs even without the randomness.
>
> # Clarity in Eq(8)
>
> >The notation in eq (8) is a bit hard to parse
>
> We appreciate that our current notation in eq(8) caused confusion; we wanted to keep our notation consistent with the other equations. Is it easier to interpret if we write equation 8 as $\hat{u}_i^{\text{actor,new}} := \hat{u}_i^{\text{actor}} + \alpha_3 \delta_i^{\text{actor}} \epsilon$?
>
> # Type of ES used in OPEN
>
> >How does the OpenAI ES strategy compare to the ES strategies used in the previous work on learned optimization, like standard ES, PES, NRES, etc?
>
> What we call "OpenAI ES" is considered "standard ES" in many circles (and is widely used in learned algorithms/learned optimizers, e.g., [1,2,3]). We use the name from evosax to prevent ambiguity. We did run preliminary experiments with CMA-ES and a genetic algorithm, but found performance insufficient.
>
> While alternatives to ES, like PES/NRES, exist, **they introduce extra bias** which can cause issues. In particular, PES and NRES update the optimizer in an *online* fashion (i.e., during the innter-training loop), and are seen as alternatives to *truncated* ES, whereas OpenAI ES completes a full inner-training loop before updating. While PES/NRES are unbiased *in theory*, their implementation of updating the meta-parameters online introduces hysteresis effects *in practice*, since the state of a system depends on its history. Given how difficult learned optimization in RL has proven historically, we chose to avoid this additional source of bias which could introduce instability into the system. We found that, in practice, OpenAI ES performed very well and we did not require truncations as training was fast enough thanks to the speed of JAX and fast convergence of OPEN compared to other methods. Thus, we decided to pursue OpenAI ES in the publication.
>
> # Conclusion
>
> We hope that we have addressed any concerns you might have had about our paper. If there is anything else we can respond to, please do let us know.
>
> ---
>
> [1] Chris Lu et al., Discovered policy optimisation. 2022.
>
> [2] Luke Metz et al., VeLO: Training Versatile Learned Optimizers by Scaling Up. 2022.
>
> [3] Luke Metz et al., Tasks, stability, architecture, and compute: Training more effective learned optimizers, and using them to train themselves. 2020.

---

> > ### Comment · Reviewer_sKXo · 2024-08-14
> >
> > Thanks for your thorough response. I will keep my positive score.

---

### Author Rebuttal · Authors · 2024-08-06

Dear reviewers,

# Thank you

We are very grateful to have received four high quality reviews, and appreciate the time and effort you all spent with our paper. We are glad that there was consistency among reviewers that our paper was easily understood and presented strong results, and that **every reviewer recommended acceptance to the conference**. There were, however, some points raised by multiple reviewers. These are addressed below as supplement to individual responses.

We have color-coded our general response for convenience:

$\color{red}{R1: sKXo}$

$\color{gold}{R2: Fr37}$

$\color{green}{R3: KVk8}$

$\color{pink}{R4: qjkS}$

# Harder Environments

Reviewers $\color{gold}{Fr37}$, $\color{green}{KVk8}$ and $\color{pink}{qjkS}$ suggested considering more complex environments than gridworlds. Therefore, we examine the generalization of OPEN to a **significantly harder environment**: Craftax-Classic [1], a JAX reimplementation of Crafter.

We consider three optimizers:
1. OPEN, learned on the gridworld distribution from section 6.4, transferring *0-shot* to Craftax-Classic. **We do not retrain OPEN for this experiment.**
2. Adam, using the hyperparameters tuned for the gridworlds in section 6.4, transferring 0-shot.
3. Adam, where we tune the learning rate in-domain and use other hyperparameters tuned for Craftax from [1]. This gives a rough idea of the ceiling on optimizer performance, as Adam is *finetuned directly on the task*.

We outline results and all hyperparameters, many of which come from [1], in the PDF. All optimizers are run on 32 seeds for $3\times10^7$ timesteps. In particular, we find:
- **OPEN drastically outperforms Adam when both generalize** from the much simpler gridworld environment distribution.
- OPEN's 0-shot transfer is only slightly worse than Adam finetuned directly in domain (expensive).

This result suggests OPEN generalizes **even better than was originally demonstrated in the paper**, and lays the foundation for truly automated RL. In particular, we show that OPEN can be trained on quick and simple environments, before transferring to more complex environments **with no additional samples or hyperparameter tuning**. We have included this result, which significantly strengthens our original claims, into our updated paper.

# Additional Algorithms

$\color{gold}{Fr37}$ asked whether OPEN worked with algorithms other than PPO. Therefore, we are *currently* training OPEN with PQN [2], a variant of DQN designed for vectorized sampling, in Asterix to explore how our approach generalizes between algorithms.

Unfortunately, due to the time taken to train a learned optimizer from scratch, and a server crash during rebuttals, our experiment has not finished within the allotted rebuttal time. We will continue to run this, and **will update OpenReview pending the conclusion of our experiment**. However, we present some initial, indicative results *from the midst of training* below.

Adam receives a return of ~36.5 (16 seeds) after finetuning, whereas OPEN *currently* achieves ~34.9 return (16 seeds). Importantly, **the meta-training performance is still consistently increasing, so we believe that OPEN will overtake Adam**. We see this as a very positive sign that OPEN is general enough for a range of RL algorithms. We will add this to our paper when the experiment has finished.

# List of Novelty

Both $\color{green}{KVk8}$ and $\color{pink}{qjkS}$ raised concerns about novelty in OPEN. However, we believe there is *significant* novelty to our work. It is worth noting that only one other work has attempted learned optimization in RL, and they fail to significantly outperform all baselines in any of their tests. On the contrary, OPEN **consistently outperforms all baselines in a range of domains** with a *completely different method*. We have consolidated a list of some of the key components introduced in OPEN, and have added some sentences to our introduction to emphasise this novelty:

- We use ES to learn to optimize for RL. This is only recently possible thanks to JAX acceleration.
- We propose a very unconstrained way to design a black-box meta-learner based on input features. This enables learning on only the ultimate goal, such as final return, rather than having handcrafted and hyperparameter-dependent regularization penalties for things such as dormancy or weight magnitude. As a result, other possible problems, such as plasticity loss, are dealt with as a byproduct. This contrasts with Optim4RL, which relies on excessive structure that limits performance.
- We highlight theoretically grounded features which can be combined to tackle a number of issues in optimization for RL. Our features are thoroughly ablated.
- We develop learnable stochasticity, a new and performant meta-learned exploration technique which parallels a learned version of parameter space noise (with some implementation details). This interacts with the different inputs of OPEN (e.g. dormancy, training proportion) to produce complex stochastic behaviour.
- We introduce a way to stabilize learned updates in domains with continuous action spaces by zero-meaning the updates, to prevent overflow issues from exponentiation. According to the Optim4RL ICLR24 rebuttals (which can be found online), this was an issue in one of their naive baselines too, and was one of the reasons they heavily constrained their update expression. Our method resolves this problem.
- We are the first example of learned optimization in RL which is able to **significantly and consistently outperform handcrafted baselines**, including beating Adam in generalization to different environments and to different architectures. This is bolstered by our additional experiments in generalization to Craftax and training with PQN.

---

[1] Michael Matthews et al., Craftax: A Lightning-Fast Benchmark for Open-Ended Reinforcement Learning. 2024.

[2] Matteo Gallici et al., Simplifying Deep Temporal Difference Learning. 2024.

---

> ### Author Response · Authors · 2024-08-12
>
> Dear reviewers,
>
> As promised in our general response, our experiment running OPEN with PQN [1] has now concluded and we can share the results. PQN is a recent version of DQN built around using vectorized sampling in JAX. Due to both compute and time constraints, we train OPEN for one seed on Asterix and evaluate against Adam. We use most hyperparameters directly from PQN, but completed an extensive finetuning on Adam covering *LR*, $\beta_1$, $\beta_2$ and *Anneal_LR*. We evaluate each hyperparameter configuration on 8 seeds.
>
> We run our evaluation of OPEN and Adam for 64 seeds, and provide standard error. Our results are as follows:
>
> | Optimizer    | Final Return ± Std Err        |
> |---------------|-------------------|
> | Adam          |   36.86 ± 0.68  |
> | OPEN    |   **40.06 ± 0.42**  |
>
> OPEN significantly outperforms Adam in this setting. While we were unable to explore different environments to Asterix, we also see no reason why these results would not transfer to the other environments explored in our paper. We will update our manuscript to include these new results.
>
> Though we can not share any additional figures due the 1 page PDF constraint, we also want to highlight that there seems to be interesting behavior of OPEN with PQN which parallels that of OPEN with PPO. In particular, performance is not uniformly better for OPEN over the training horizon, and OPEN only begins to outperform Adam from about halfway through training. We will include the return curve into our appendix, like with all our other experiments, to demonstrate the occurence of this effect. While additional analysis would be interesting in this setting, to see if the behavior of OPEN is consistent across algorithms, we have not had the time within the rebuttal period to carry this out.
>
> We hope that the reviewers find this additional result interesting, and are happy to answer any extra queries about the performance of OPEN with other algorithms.
>
> [1] Matteo Gallici et al., Simplifying Deep Temporal Difference Learning. 2024.

---

### Decision · Program_Chairs · 2024-09-25

**Decision:**

Accept (spotlight)

**Comment:**

The paper proposes an interesting research direction to tackle the specific issues that arise in deep RL optimization. The approach suggests to make use of a "learned optimization" to overcome these problems in a meta-learning setting.

Here are the strong points of the paper provided in the reviews:
- The contributions has the potential to lead to follow-up research with a significant impact (all reviewers).
- The strong evaluation and extensive ablation studies are convincing and support well the goal and claims of the paper (reviewers Fr37 and KVk8).
- Discussion of acceleration via parallelization in JAX is appreciated (reviewer sKXo).
- The quality of the paper is appreciated in terms of writing concerning introduction, related work, analysis of results, etc. (all reviewers).

A few (minor) weaknesses or suggestions:
- The idea about meta-learning an optimizer is not fully original (see reviewer KVk8).
- Even more extensive experiments could strengthen the results even further (see reviewer Fr37) and some additional experiments were suggested (e.g. reviewers sKXo and qjkS).

**Summary**: Given the strong quality of the paper, we recommend the paper for acceptance and even a spotlight. The originality of the paper is good but also not highly significant and we don't recommend the paper for an oral presentation.